# Genome-wide association study identifies a gene responsible for temperature-dependent rice germination

Hideki Yoshida [1,2,5] ✉, Ko Hirano[2,5], Kenji Yano[2,3,5], Fanmiao Wang[2], Masaki Mori[2], Mayuko Kawamura[2], Eriko Koketsu[2], Masako Hattori[2], Reynante Lacsamana Ordonio[2], Peng Huang[2], Eiji Yamamoto[4] & Makoto Matsuoka [1,2] ✉

Environment is an important determinant of agricultural productivity; therefore, crops have been bred with traits adapted to their environment. It is assumed that the physiology of seed germination is optimised for various climatic conditions. Here, to understand the genetic basis underlying seed germination, we conduct a genome-wide association study considering genotype-by-environment interactions on the germination rate of Japanese rice cultivars under different temperature conditions. We find that a 4 bp InDel in one of the 14-3-3 family genes, *GF14h*, preferentially changes the germination rate of rice under optimum temperature conditions. The GF14h protein constitutes a transcriptional regulatory module with a bZIP-type transcription factor, OREB1, and a florigen-like protein, MOTHER OF FT AND TFL 2, to control the germination rate by regulating abscisic acid (ABA)-responsive genes. The GF14h loss-of-function allele enhances ABA signalling and reduces the germination rate. This allele is found in rice varieties grown in the northern area and in modern cultivars of Japan and China, suggesting that it contributes to the geographical adaptation of rice. This study demonstrates the complicated molecular system involved in the regulation of seed germination in response to temperature, which has allowed rice to be grown in various geographical locations.

The Japanese archipelago is over 3000 km long from north (latitude: 45°) to south (20°) with highly varied topography and climatic conditions. Rice (*Oryza sativa*) is grown in most areas in Japan, and widespread rice cultivation relies on the genetic improvement of cultivars to adapt to diverse climatic conditions.

For example, heading time is one of the most important traits for rice plants to adapt to climate change. Using a genome-wide association study (GWAS) with Japanese cultivars, we identified more than ten loci responsible for heading[1]. Further studies revealed that some

specific alleles of heading genes were preferentially localised in specific areas of Japan[2], indicating that breeders selected these alleles for regional adaptation, especially temperature conditions. COLD1, which can confer cold tolerance to rice, demonstrates the underlying genetic factors involved in the expansion of the rice-growing area to the northern part of East Asia[3].

Seed germination is highly plastic and can be modulated by environmental cues, such as temperature, light, and soil conditions[4]. In general, seed germination of rice is attained in a few days at 27–37 °C,

[1]Institute of Fermentation Sciences, Fukushima University, Fukushima 960-1248, Japan. [2]Bioscience and Biotechnology Center, Nagoya University, Aichi 464-8601, Japan. [3]Statistical Genetics Team, RIKEN Center for Advanced Intelligence Project, Tokyo 103-0027, Japan. [4]Graduate School of Agriculture, Meiji University 1-1-1 Higashi-Mita, Tama-ku, Kawasaki, Kanagawa 214-8571, Japan. [5]These authors contributed equally: Hideki Yoshida, Ko Hirano, Kenji Yano. ✉e-mail: hyoshida@agri.fukushima-u.ac.jp; matsuoka@agri.fukushima-u.ac.jp

and the temperature conditions below or above this range substantially disrupt germination[5]. Therefore, temperature-dependent seed germination might have been an important trait for widespread rice cultivation in Japan, and Japanese varieties may show genetic diversity for this trait.

14-3-3 proteins are highly conserved proteins, widespread in eukaryotic organisms[6]. Among eukaryotes, plants have the largest number of 14-3-3 genes, with 15 in *Arabidopsis* and 8 in rice[7]. The protein family members are classified according to their amino acid sequence similarities into two distinct groups: the ε and the non-ε group[7]. A common trait of 14-3-3 is their ability to bind to target proteins through the recognition of phosphorylated consensus motifs[7]. Depending on the phosphorylated target, association of 14-3-3 proteins can have different functional consequences, leading to regulation of its enzymatic activity, subcellular localization, protein stability or alteration of protein-protein interactions[7]. At present, a wide range of 14-3-3 interactants playing a pivotal role in various physiological processes, such as growth and development and stress response, have been identified[7]. Additionally, a growing body of evidence has emerged regarding the involvement of 14-3-3 proteins as key players in different aspects of plant hormone physiology[7].

In this work, we conduct a GWAS to identify the genetic mechanism modulating seed germination under different temperatures. For this, we use an updated GWAS system with a genotype-by-environment (G × E) interaction (G × E GWAS)[8] and find a causal gene for temperature-dependent seed germination, encoding a 14-3-3 protein, GF14h. We further investigate the mechanism of GF14h mediated temperature-dependent germination of rice. Moreover, the results allow us to further explain the success of domesticated rice over a large geographical region.

## Results

### Detection of a G × E gene controlling seed germination

To identify genes regulating seed germination under specific temperature conditions, the germination rates of 164 Japanese rice (*O. sativa* subsp. *japonica*) varieties were observed under two different immersion temperatures, 15 °C for 96 h and 30 °C for 24 h (Supplementary Data 1 and Supplementary Figs. 1, 2a–c). G × E GWAS was conducted using a linear mixed model with four terms: genotype (G), environment (E), G × E, and residual error with correction for kinship bias (see Methods). A peak at 23.5 Mb in chromosome (Chr.) 11 was denoted as Peak 1 (red arrow in Fig. 1a; Supplementary Data 2 and Supplementary Fig. 3a). We assumed Peak 1 to be a good candidate for the G × E locus, and further evaluated the effect of Peak 1 by simple

GWAS. We further performed GWAS on germination rates at 15 °C and 30 °C. Narrow-sense heritability was 51.5% and 53.2% for 30 °C and 15 °C, respectively. Peak 1 was detected as the only significant peak at 30 °C (Fig. 1b and Supplementary Fig. 3b). At 15 °C, no significant peak was found, whereas the -log10(*P*-value) of Peak 1 was decreased to approximately 2.6 (Fig. 1c and Supplementary Fig. 3c), implying that Peak 1 is mainly associated with the germination rate at 30 °C.

Previous studies have reported a major quantitative trait locus (QTL), *qLTG3*, which dominantly controls germination under low temperature in Japanese rice varieties[9]. However, our GWAS at 15 °C did not detect this QTL (Fig. 1c). Three *qLTG3* alleles with different functions were present in our GWAS panel: haplotype (Hap.)1 (partial loss-of-function [LOF]), 2 (LOF), and 3 (gain-of function [GOF]) (Supplementary Fig. 4a). The most of GWAS platforms are designed to analyse bi-allelic variants, and therefore, allelic heterogeneity[10] are ignored or forcibly converted into bi-allelic state (e.g., reference type and another major allele)[11]. We speculated that these functionally different alleles of *qLTG3* may decrease the statistical power of the GWAS. In the variant list, these polymorphisms were regarded as allelic heterogeneity and neglected for GWAS. To resolve this problem, we converted the tri-allelic variants to three bi-allelic variants for GWAS (Supplementary Fig. 4) and detected *qLTG3* as a high peak at 15 °C but not for G × E GWAS (Fig. 1d–f, Supplementary Fig. 3d–f and Supplementary Data 3). These results indicate that Peak 1 can mainly explain the difference in temperature-dependent germination among the varieties we studied.

We focused on Peak 1 for further analyses. The candidate region of Peak 1 included 8 polymorphisms with significant *P*-values and amino acid exchange or frame shift mutations (Fig. 2, Supplementary Fig. 5 and Supplementary Table 1). When evaluating the functional impact of these mutations (Supplementary Figs. 6 and 7), we found a 4 bp InDel in the coding sequence of LOC_Os11g39540/Os11g0609600, which was annotated as a 14-3-3 protein, GF14h (Fig. 2b)[12]. As the reference genome of Nipponbare (NPB, referred to as GF14h^Hap.1) contains a 4 bp deletion, leading to a lack of the conserved region among its orthologs in various plants (Supplementary Fig. 7), GF14h^Hap.1 could be a LOF allele, whereas GF14h^Hap.2 and GF14h^Hap.3 could be functional because their sequences are shared by the orthologs (Supplementary Fig. 7). We confirmed that varieties carrying GF14h^Hap.1 showed germination rates lower than GF14h^Hap.2/3 plants at 30 °C (Fig. 2c), while they were almost similar at 15 °C (Fig. 2d). Phylogenetic analysis revealed that GF14h is a grass-specific ε-type GF14 (yellow box in Supplementary Fig. 8). We examined its organ expression pattern using transgenic plants carrying the GF14h^Hap.2 promoter::GUS and found that GF14h^Hap.2 was expressed in

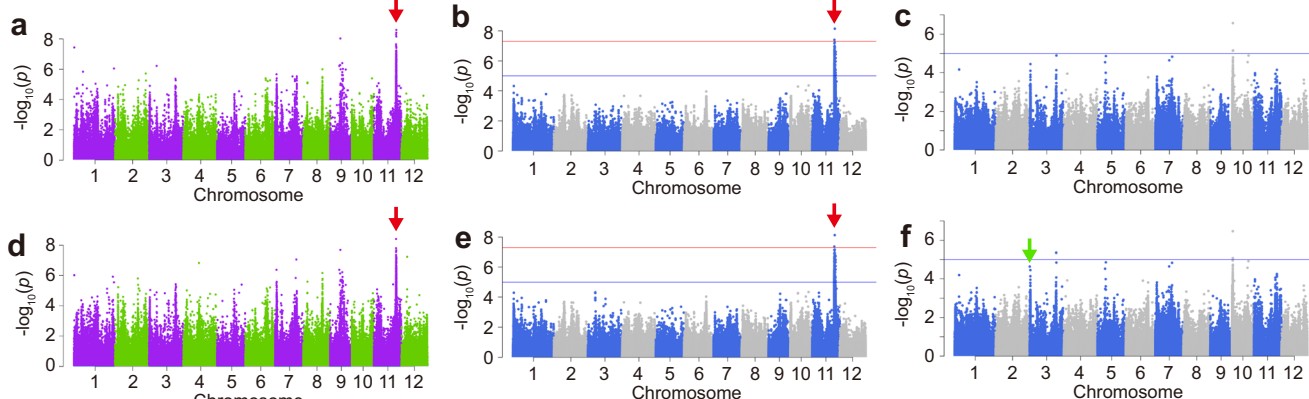

**Fig. 1 | Genome-wide association study for rice germination rate. a** Genotype × environment (G × E) genome-wide association study (GWAS) for germination rate at 30 °C for 24 h vs 15 °C for 96 h. GWAS for germination rate at 30 °C for 24 h (**b**) and 15 °C for 96 h (**c**). **d** G × E GWAS with the modified variant list. GWAS at 30 °C for 24 h (**e**) and 15 °C for 96 h (**f**) with the modified list. Horizontal red lines indicate 5% genome-wide significance threshold after Bonferroni-correction. Blue lines indicate -log10 *P* values = 5. Peak 1 and Peak 2 are shown by red and green arrows, respectively.

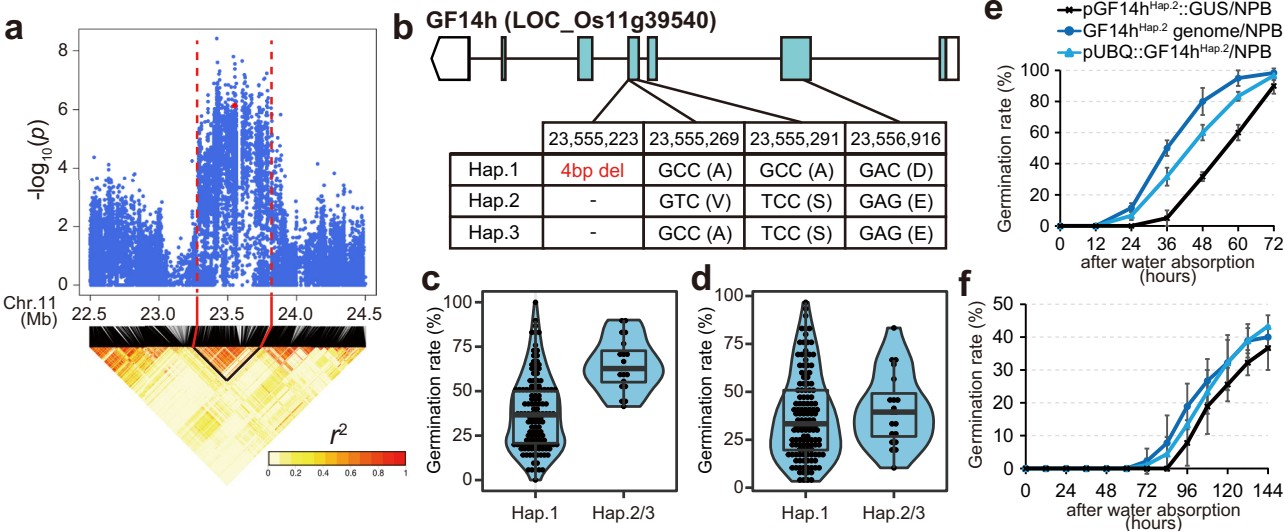

**Fig. 2 | Identification of the causal gene for G × E Peak 1. a** Local Manhattan plot (top) and LD heatmap (bottom) surrounding Peak 1. Red dashed lines indicate the close up of Peak 1 region shown in Fig. 1d, and the red dot indicates the position of the 4 bp InDel variation in GF14h. **b** Exon-intron structure of *GF14h* with poly-morphisms inducing amino acid exchanges. Violin plots showing germination rate of cultivars that have loss-of-function (haplotype [Hap.]1) and gain-of-function (Hap.2/3) at 30 °C for 24 h (**c**) and 15 °C for 96 h (**d**). Haplotype information is shown in Supplementary Data 1. *n* = 131 (Hap.1), 20 (Hap.2). Edges of box indicate 25 and 75 percentile points along with medians. Whiskers indicate minima and maxima. Complementation of GF14h function by introduction of the Hap.2 genome or ubiquitin promoter (pUBQ)::GF14h$^{Hap.2}$ CDS at 30 °C (**e**) and 15 °C (**f**). The plants introduced with the promoter of GF14h$^{Hap.2}$ (p GF14h$^{Hap.2}$)::GUS were used as a control. The centre for the error bars represents mean. Error bars, s.d. (*n* = 3 bio-logically independent samples). Source data are provided as a Source Data file.

vascular tissues of stems, anthers, roots, aleurone of embryos, and calli (Supplementary Fig. 9), consistent with the public data derived from NPB carrying its LOF allele (Supplementary Fig. 10a–c). We also performed a complementation analysis using NPB as a background, and confirmed that the introduction of the GF14h$^{Hap.2}$ genome or ubiquitin promoter::GF14h$^{Hap.2}$ coding sequence increased the ger-mination rate at 30 °C, whereas only a slight effect was observed at 15 °C (Fig. 2e, f). These results confirm that the effect of GF14h on the germination rate depends on G × E.

### GF14h is a negative regulator in ABA signalling

To elucidate the biological function of GF14h, we studied its invol-vement in the ABA signalling pathway, which is important for seed germination and public data shows that the expression of GF14h is regulated by ABA in roots (Supplementary Fig. 10d, e). First, we compared the ABA responsiveness of germination rate between seeds with or without functional GF14h and found that functional *GF14h* dramatically reduced ABA responsiveness (Fig. 3a) and expression of three ABA responding genes, *OsRab16A*, *OsLea3*, and *OsEM* (Fig. 3b). Since direct interactions between GF14s and bZIP factors have been reported in various biological processes, such as flowering and tuberization[13,14], we next examined the interaction of GF14h with bZIP proteins, OREB1/OsABI5 and TRAB1/OsbZIP66, which have been reported as key factors for ABA-dependent responses, such as inhibition of germination and stress response[15–17]. Using a yeast two-hybrid system (Y2H), we observed that the GF14h$^{Hap.2}$ product established active interactions with OREB1, low interactions with TRAB1, and barely any interaction with the GF14h$^{Hap.1}$ product (Fig. 3c). Thus, we focused on the relationship between GF14h and OREB1. In rice mesophyll protoplasts transiently expressing GFP-GF14h$^{Hap.1}$ and GFP-GF14h$^{Hap.2}$, the GFP-GF14h$^{Hap.1}$ and GFP- GF14h$^{Hap.2}$ were mainly observed in the endoplasmic reticulum (ER) and nucleus and cytosol, respectively (Fig. 3d and Supplemen-tary Fig. 11a–d), while GFP-OREB1 was localised in the nucleus (Fig. 3d). We also confirmed that GF14h$^{Hap.2}$ and not GF14h$^{Hap.1}$ inter-acted with OREB1 in the plant nucleus using bimolecular florescence complementation (BiFC) assay (Fig. 3e and Supplementary Fig. 11e).

Furthermore, we focused on the molecular relationship between GF14h$^{Hap.2}$ and OREB1. Considering the temperature-dependent function of GF14h, we analysed the effect of temperature on the interaction. Additional BiFC assays with incubation at different temperatures showed that temperature did not substantially affect the physical interaction of GF14h$^{Hap.2}$ and OREB1 (Supplementary Fig. 11f, g). This interaction depending on the conserved C-terminus region of GF14h in nucleus mimics the interaction between the GF14 florigen receptor and bZIP-type FD in the Flowering Activation Complex (FAC)[13]. Thus, we replaced S385 in OREB1, which corre-sponds to the phosphorylation site essential for the OREB1 and GF14 protein interaction[18], with Ala (S385A) or Glu (S385E) to mimic dephosphorylation and phosphorylation, respectively, in order to examine the role of its phosphorylation state. S385A attenuated its interaction without relocation of OREB1, while S385E did not obviously change its interaction or localisation (Fig. 3d, e). This observation was quantitatively confirmed by three independent large-scale assays (Fig. 3f). These results demonstrated that GF14h and OREB1 form a complex in the nucleus partially through the phosphorylation of S385, similar to that of FAC.

We also examined the involvement of the third homologous component of the FAC, MOTHER OF FT AND TFL (MFT), which has been reported as another component that regulates seed germination in wheat (TaMFT)[19]. There are two MFTs in rice, MFT1 and MFT2; MFT2 corresponds to TaMFT (Supplementary Fig. 12). In fact, the embryo-specific expression and ABA-inducibility of MFT2 suggested its role in the regulation of seed germination (Supplementary Fig. 13). Further-more, MFT2 defective plants generated using CRIPSR-Cas9 were much less sensitive to ABA than controls (Supplementary Fig. 14). GFP-MFT2 was observed in the nucleus and cytosol (Fig. 4a), whereas the BiFC signal of GF14h$^{Hap.2}$-MFT2 was observed mainly in the ER similar to GF14h$^{Hap.1}$ (Fig. 4b and Supplementary Fig. 11a–d). GF14h$^{Hap.1}$ could not interact with MFT2 (Supplementary Fig. 11e). OREB1-MFT2 was speci-fically localised in the nucleus. S385A in OREB1 impeded the interac-tion with MFT2, whereas no effect of S385E was found (Fig. 4b, c). Although there was no clear interaction of GF14h-MFT2 or OREB1-MFT2 with Y2H (Supplementary Fig. 15), such a discrepancy between

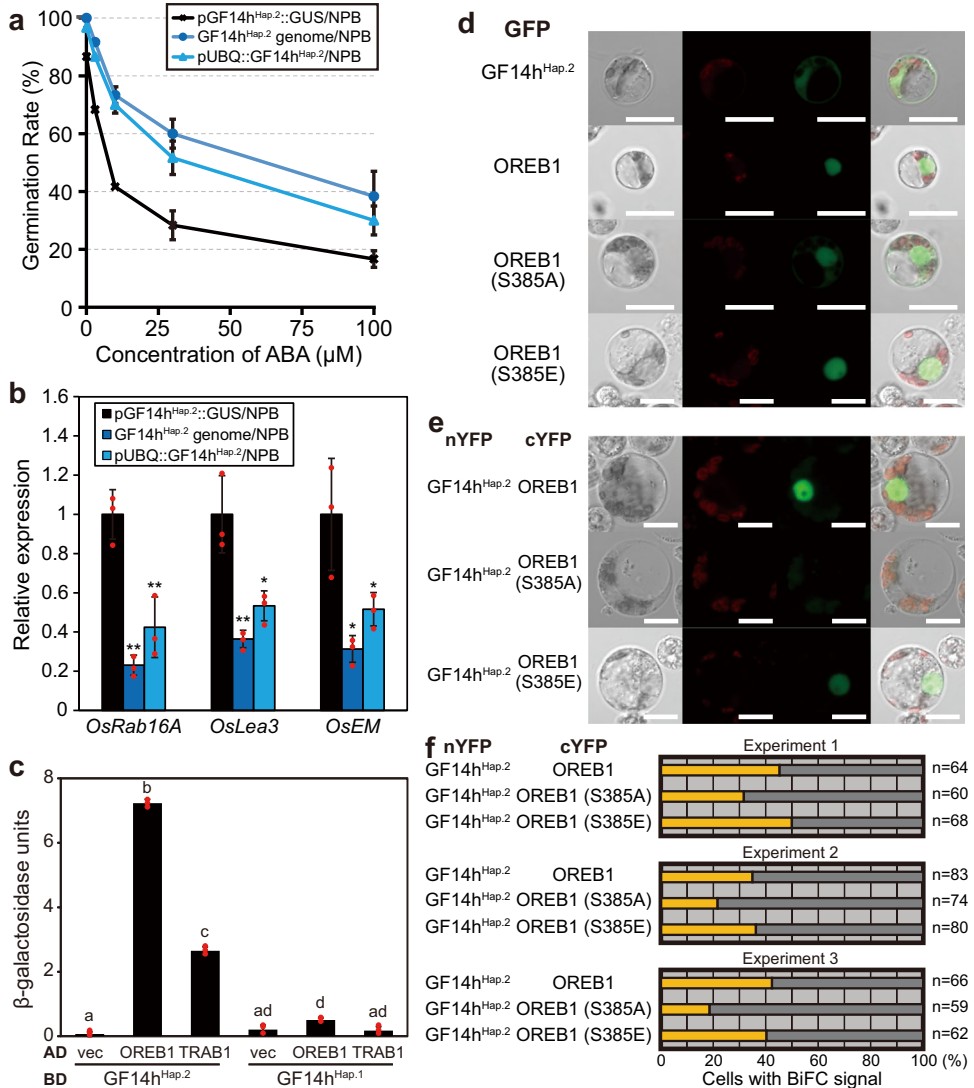

**Fig. 3 | GF14h is involved in the ABA signalling pathway. a** ABA responsiveness of the seed germinability of the GF14h complemented or control plants for 72 h at 30 °C. Error bars, s.d. (*n* = 3 biologically independent samples). **b** Relative transcription of ABA-responsive genes in seeds at 12 h after imbibition at 30 °C. Error bars, s.d. (*n* = 3 biologically independent samples). The relative expression in pGF14h$^{Hap.2}$::GUS plants (left) was set as 1. **$P$ value < 0.01, *$P$ value < 0.05 based on two-sided student's *t* test. The exact $P$ values were shown in Source Data file. **c** Yeast two hybrid system for the interaction of OREB1 or TRAB1 and GF14h haplotype [Hap.]1 or Hap.2 product. Letters indicate significant differences (*n* = 3 biologically independent samples, $P$ < 0.01, Tukey's HSD test). The exact $P$ values were shown in Source Data file. **d** Subcellular localisation of GFP-GF14h$^{Hap.2}$, and the native and mutagenised GFP-OREB1 in rice mesophyll protoplast. Individual and merged images of differential interference contrast images of protoplasts YFP (green) and chlorophyll autofluorescence (red) are shown. Scale bars, 10 μm. Each experiment was repeated independently for at least 3 times with similar results. **e** BiFC assay for the GF14h$^{Hap.2}$-OREB1 interaction. Scale bars, 10 μm. **f** Quantification of the results in (**e**). Percentage of cells showing BiFC signal in nuclei is indicated by yellow bars with the total number of cells observed on the right. Each experiment was repeated independently for at least 3 times with similar results. Source data are provided as a Source Data file.

in vivo and in yeast suggests that other factors in rice cells could be essential for their interaction.

Next, we investigated the role of GF14h in ABA signalling by a transient assay using a reporter gene, firefly luciferase (fLUC), under the control of the *OsEM* promoter, which is often used as a reporter in ABA signalling in rice[20]. Its expression was enhanced by OREB1, whereas this enhanced expression was partially diminished by GF14h$^{Hap.2}$ (Fig. 4d). ABA significantly enhanced the transactivation activity of OREB1, and the suppression by GF14h$^{Hap.2}$ also occurred in a dose-dependent manner, while the coexistence of MFT2 eased the suppressive effect of GF14h$^{Hap.2}$ in a dose-dependent manner (Fig. 4d). This negative effect of GF14h$^{Hap.2}$ and positive effect of MFT2 were not observed in the presence of OREB1 (S385A) (Fig. 4e) or absence of OREB1 (Supplementary Fig. 16a). In addition, GF14h$^{Hap.1}$ could not repress the activity of OREB1 and diminish the effect of MFT2

(Supplementary Fig. 16b). These results indicated that the effects of GF14h$^{Hap.2}$ and MFT2 depend on their interaction with OREB1. We further conducted BiFC experiments using non-tagged third factors. There was no change in the signal of GF14h$^{Hap.2}$-OREB1 or OREB1-MFT2 in terms of localisation or intensity when MFT2 or GF14h$^{Hap.2}$ was present, respectively (Fig. 4f, g and Supplementary Fig. 16c). In contrast, the localisation of GF14h$^{Hap.2}$-MFT2 changed to the nucleus in the presence of OREB1, whereas such relocation was rarely observed when OREB1 (S385A) was present (Fig. 4f, g). These results indicate that the triple complex formation preferentially occurs in the nucleus via the phosphorylated OREB1. A co-immunoprecipitation (co-IP) assay confirmed the GF14h$^{Hap.2}$-OREB1 interaction in vivo, and that this interaction was unaffected by the presence of MFT2 (Fig. 4h and Supplementary Fig. 16d), indicating that MFT2 is not a competitor against their formation but as a partner of a ternary complex in the

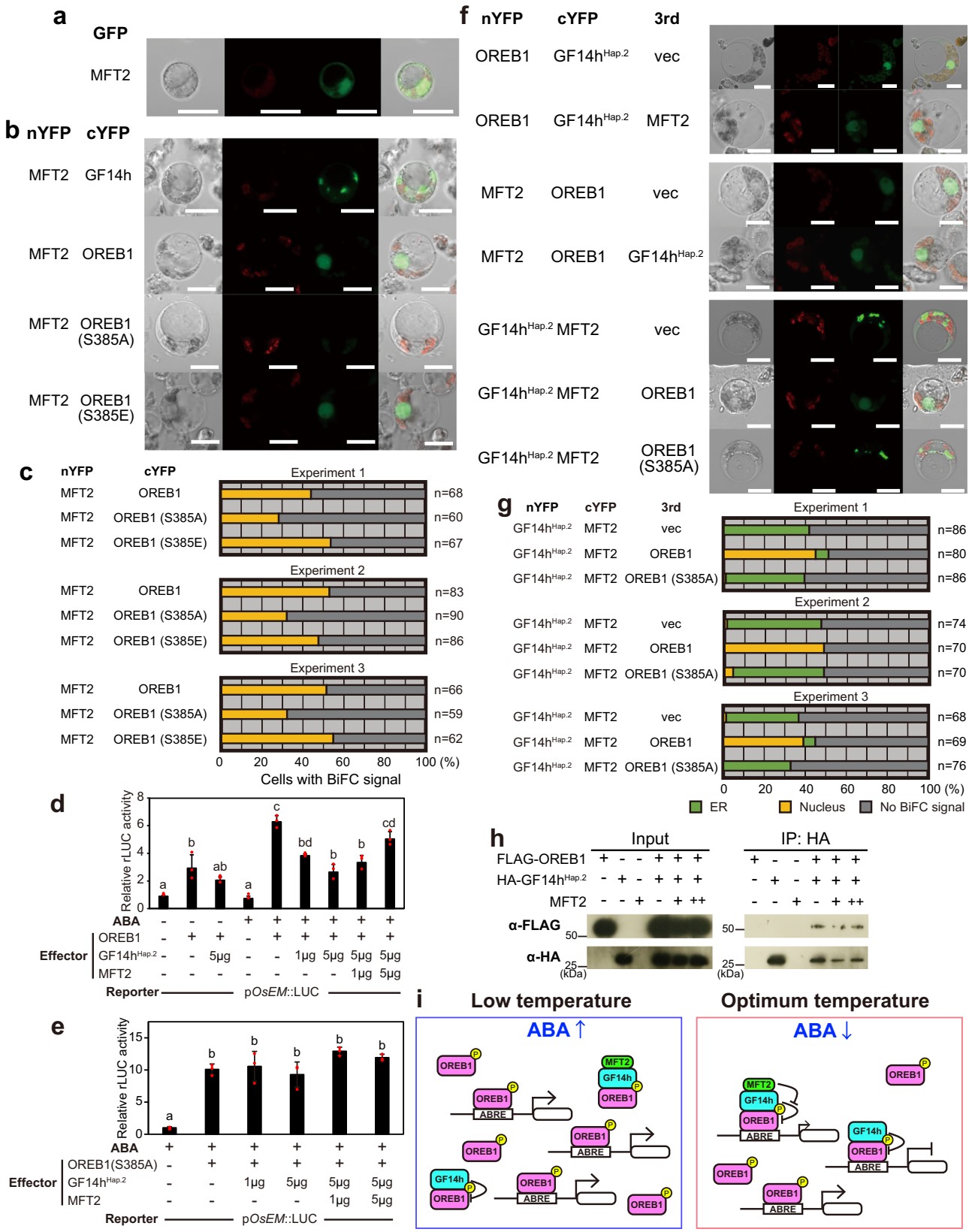

nucleus. Complying with this, the amount of OREB1 protein was not affected by that of GF14h[Hap.2] (Supplementary Fig. 16e, f). GF14h and MFT2 are thought to function as a suppressor and a de-suppressor, respectively, against the OREB1 function by forming a dimer (GF14h-OREB1) or trimer (GF14h-OREB1-MFT2) for fine-tuning of seed germination through regulation of ABA signalling (Fig. 4i, see Discussion).

## Distribution and transition of LOFs of GF14h and qLTG3 during rice breeding

We analysed the haplotype frequency of *GF14h* and *qLTG3* among various rice subpopulations using the 3010 *O. sativa* accessions (3 K panel)[21] and our own *O. rufipogon* database (Supplementary Table 2). We found five additive haplotypes in *O. sativa*, Hap.4–Hap.8, and six in

**Fig. 4 | The transcriptional module comprised of GF14h, OREB1, and MFT2 for ABA-mediated seed germination. a** Subcellular localisation of GFP-MFT2. Scale bars, 10 μm. Each experiment was repeated independently for at least 3 times with similar results. **b** BiFC assay for interaction of MFT2 with GF14h[Hap.2] or OREB1. Each experiment was repeated independently for at least 3 times with similar results. Scale bars, 10 μm. **c** Quantification of BiFC assay in (**b**). **d** Transient reporter assay for testing the transactivation effects of OREB1, GF14h[Hap.2], and MFT2 on the expression of *OsEM::rLUC* with/without ABA in rice mesophyll protoplasts. The activity by vector control (far left bar) was set as 1. **e** The same experiment as (**d**) using OREB1 (S385A). In **d** and **e**, letters indicate significant differences (*P* < 0.01, Tukey's HSD test, *n* = 3). Error bars, s.d. The exact *P* values were shown in Source Data file. **f** BiFC assay of interaction among GF14h, OREB1[Hap.2], and MFT2 with the non-tagged 3rd proteins. Each experiment was repeated independently for at least 3 times with similar results. Scale bars, 10 μm. **g** Quantification of re-localisation of GF14h-MFT2 signal by OREB1 in (**f**), whereas that for other combinations are shown in Supplementary Fig. 16b. **h** Co-immunoprecipitation of OREB1 by GF14h[Hap.2]. OREB1 and GF14h[Hap.2] were tagged with FLAG and haemagglutinin (HA), respectively. Gel blots were probed with anti-FLAG or anti-HA antibody. The uncropped image is shown in Supplementary Fig. 16d. Each experiment was repeated independently for at least 3 times with similar results. **i** Model of the transcriptional regulation by the GF14h, OREB1, and MFT2 module with respect to temperature-dependent seed germination. Source data are provided as a Source Data file.

*O. rufipogon*, RUF1–6 (Supplementary Table 3). Haplotype network analysis predicted that Hap.3 is the ancestral haplotype (Supplementary Fig. 17a) and the others are derived haplotypes. Lacking the whole genome and half of the 3′-terminal regions (Supplementary Table 3), the origin of Hap.7 and Hap.8 were predicted by their surrounding sequence homology. Among the derived haplotypes, Hap.1 (4 bp deletion), Hap.6 (1 bp insertion), Hap.7 (genome deletion), and Hap.8 (genome deletion) were predicted as LOFs, while others could maintain their function, because their exchanged residues were variable among monocot plants (Supplementary Fig. 7). Interestingly, these LOFs are subpopulation-specific, that is, Hap.1 and Hap.8 are preferentially found in temperate (GJ-tmp) and tropical (GJ-trp) *O. japonica*, whereas Hap.6 and Hap.7 are found in *O. indica* XI-3 and XI-1A, respectively (Fig. 5a and Supplementary Fig. 17b), suggesting that these LOFs were independently selected after emergence of these subpopulations. All *O. rufipogon* accessions carry functional haplotypes with two exceptions in Or-I with Hap.7 (Fig. 5a), indicating that this haplotype originated from wild rice.

We found regional occurrences of these LOF haplotypes: Hap.1 is preferentially present in rice from East Asian and European countries, Hap.6 is mainly in Indochina except Malaya, Hap.7 is dominant in China, and Hap.8 in Indonesia (Fig. 5b). We calculated the frequency of these LOFs within each dominant subpopulation in each area (Fig. 5c) and confirmed preferential localisation of Hap.6, Hap.7, and Hap.8 in Indochina, China, and Indonesia, whereas Hap.1 also showed significant local preference in East Asia than in Europe within GJ-tmp. These observations indicate that LOFs were independently selected in each subpopulation and the local area. The same analysis was performed for *qLTG3*. There were four haplotypes in the 3 K and *O. rufipogon* panels (Supplementary Table 4 and Fig. 5d). The LOF haplotypes, Hap.1 (partial LOF) and Hap.2 (LOF)[22], were exclusively found in GJ-tmp, indicating that the LOFs were specifically selected in GJ-tmp. Furthermore, the frequency of LOF in East Asia was significantly higher than that in Europe (Fig. 5e), indicating that slow germination haplotypes of *qLTG3* were preferentially selected in East Asia relative to Europe, similar to *GF14h*.

Thereafter, we checked whether these seed germination genes had been used for the northward expansion of the rice growing area. We compared the haplotype frequency between Japanese landraces and varieties established before and after 1990 in the northern and southern areas of Japan. For *qLTG3*, transition occurred from GOF (Hap.3) to partial LOF (Hap.1) and then to LOF (Hap.2) in the process of breeding, whereas its transition occurred more rapidly and thoroughly in the northern area than in the southern area (Fig. 5g). As for *GF14h*, a similar trend was also observed (Fig. 5f), although the transition was not as drastic as that of *qLTG3*, especially when compared between the northern and southern regions. We also performed the same analysis using Chinese temperate *japonica* in the 3 K panel and found the same trend as above but clearer (Supplementary Fig. 18a, b).

## Discussion

Studies on G × E interactions have long been considered useful not only for crop science[23,24] but also for investigating biological pathways[5]. Statistical methods have been proposed based on the concept that G × E interactions can be predicted from genomic and environmental covariates[25,26]. However, the genetic basis of G × E at the gene level has seldom been successfully studied[27,28]. For example, G × E interactions have been reported to influence dormancy and germination plasticity in *A. thaliana*[29,30]; however, genes involved in such G × E interactions have not been isolated. In this study, we performed G × E GWAS to isolate genes involved in temperature-dependent seed germination in rice. Since G × E GWAS is a developing research area, various methods have been proposed (e.g., Moore et al.[31], Dahl et al.[32]). Among these, we applied the method proposed in Yamamoto and Matsunaga[33] as the experimental design of this study completely fits the assumption of our method. This analysis identified the region on Chr. 11 containing the gene(s) responsible for temperature-dependent germination. The following physiological and biochemical analyses confirmed *GF14h* as, at least, one of the G × E genes, preferentially functioning at 30 °C but not at 15 °C. The previous study has demonstrated that low temperature increases the content of ABA in plumule during germination and induces the expression of *OREB1*[17]. Therefore, it could be a reasonable explanation for the physiological function of GF14h that higher ABA content at 15 °C raise the amount of OREB1 and may mask the transcriptional effects of GF14h. On the other hand, at optimum temperatures, low levels of ABA and OREB1 would allow GF14h to influence germination (Fig. 4i).

Conversely, *qLTG3*, which has been reported to regulate seed germination under low temperature[9], was not detected in our GWAS at 15 °C (Fig. 1c), possibly due to its multi-allelic variant. Several methods have been proposed to analyse the correlation between multiple-allelic functional polymorphisms and phenotypes[34,35]. In this study, we applied an approach which converts tri-allelic variants into three bi-allelic variants (see details in Methods and Supplementary Fig. 4b) which enabled us to find *qLTG3* as a high peak at 15 °C (Fig. 1f). Additionally, *qLTG3* was not detected in the G × E GWAS, indicating that *qLTG3* cannot be regarded as a G × E locus under the present conditions.

This study revealed that GF14h is involved in germination by regulating ABA signalling. GF14h suppresses ABA signalling by interacting with OREB1 in the nucleus decreasing its transcriptional activity, thereby releasing seed dormancy. Furthermore, MFT2 forms a ternary complex with OREB1-GF14h in nuclei and attenuates seed germination by de-suppressing OREB1 suppression by GF14h. MFT2 may functions to prevent GF14h from recruiting other co-repressor(s) or to recruit additional transcriptional activation factor(s) in nuclei.

Although the involvement of GF14 proteins in seed germination has been reported, our study suggests a different molecular mechanism. Schoonheim et al.[36] reported that GF14s are positive factors of ABA-inducible transcription machinery in seed germination. However, our results demonstrated that GF14h increased seed germination by suppressing ABA signalling (Figs. 2e, 3a, b). This may be because all the barley GF14s studied by Schoonheim et al.[36] are non-ε-type GF14s, whereas GF14h is an ε-type GF14, and different types of GF14s might have antagonistic functions. Moreover, the exact function of non-ε-type GF14s in ABA signalling requires further investigation. The

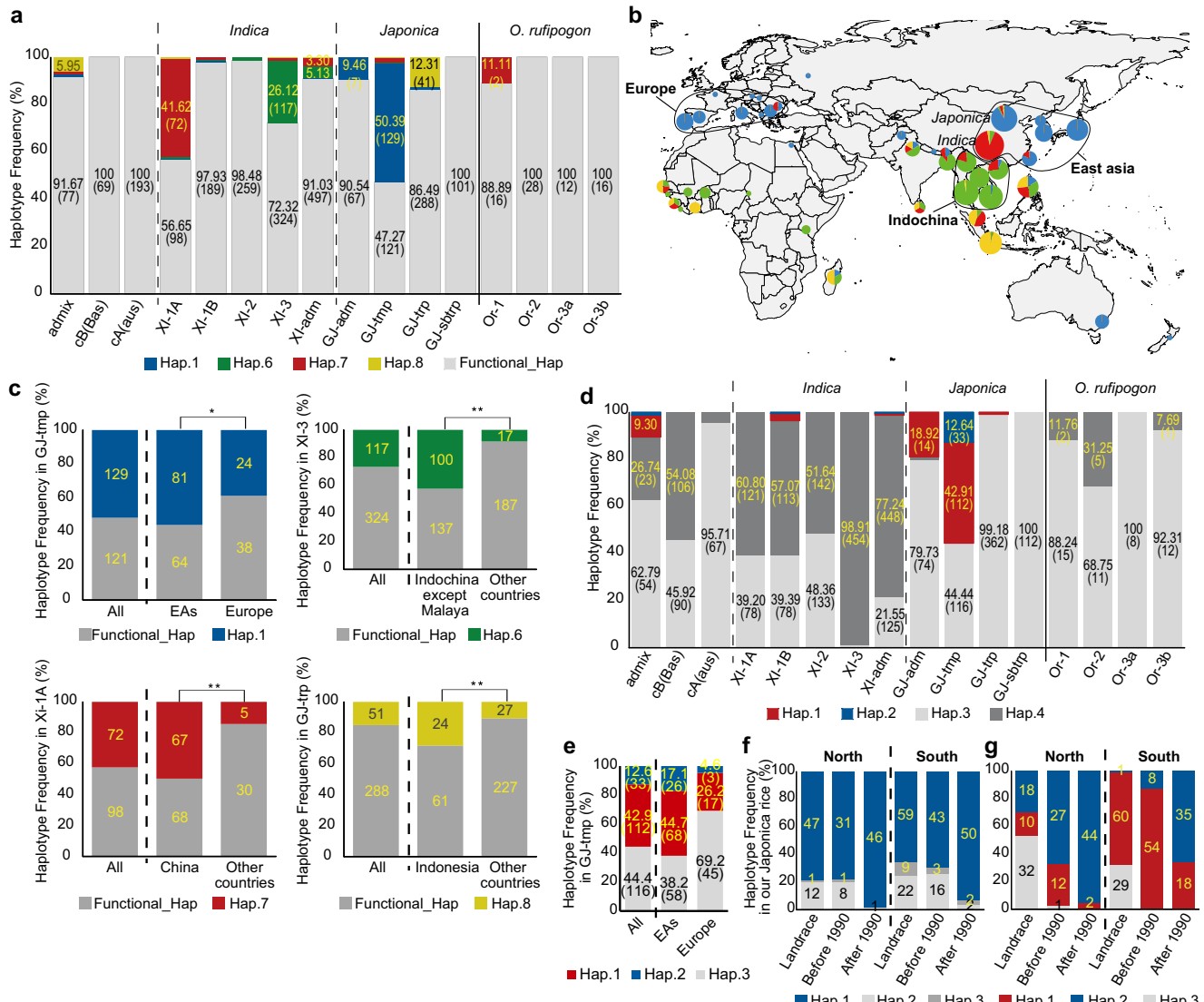

**Fig. 5 | Distribution and transition of loss-of-function (LOF) haplotypes of GF14h and qLTG3. a** Haplotype frequency of GF14h among subpopulations of domesticated and wild rice. Four LOF haplotypes are present, while all functional haplotypes are combined as "Functional_Hap". Frequency of all haplotypes is shown in Supplementary Fig. 17b. Numbers in bars represent percentage and the number of accessions in parentheses. Hap.7 and Hap.8 were confirmed by directly checking the alignment of reads by the Integrative Genomics Viewer (https://software.broadinstitute.org/software/igv/). **b** Geographical distribution of the GF14h LOF haplotypes. Ratio of each LOF to

the total LOFs was calculated in each country. In China, it was separately calculated in *japonica* and *indica*, whereas there was no separation in other countries. The colour is the same in **a**. **c** Haplotype frequency of four GF14h LOF haplotypes in each area. * and ** represent *P* value < 0.05 or 0.01, respectively. The exact *P* values were shown in Source Data file. **d** Haplotype frequency of *qLTG3*. **e** Geographical distribution of *qLTG3* haplotypes in temperate *japonica*. Haplotype frequency of *GF14h* (**f**) and *qLTG3* (**g**) in Japanese *japonica* landrace, modern varieties before and after 1990 grown in northern and southern areas of Japan.

---

involvement of MFTs in ABA signalling and seed germination is also controversial. Xi et al.[37] reported that MFT promotes embryo growth and enhances germination of *Arabidopsis* seeds by constituting a negative feedback loop in ABA signalling. However, a recent study reported that rice MFT2 can delay seed germination by interacting with bZIP factors via ABA signalling[38], which supports our model that MFT2 functions as a positive regulator of ABA signalling by forming a complex with GF14h and OREB1.

Interestingly, the complex involved in seed germination is composed of the same family of proteins as the FAC complex, GF14b/c, FD1, and Hd3a/RCNs, which regulate flowering[13,39]. Both complexes are localised in the nucleus and function as transcriptional control complexes with bZIP proteins. Additionally, differences can be observed in their molecular biological features. For example, the interaction between each FAC component occurs in yeast cells[13], but not in the

GF14h-OREB1-MFT2 complex, suggesting that this complex is less uniform than that of FAC. Indeed, OREB1 by itself can transactivate its target, whereas FD1 requires other components[13]. It is possible that the GF14h-OREB1-MFT2 complex is a part of a larger regulatory module, which is consistent with the result that ABA upregulated the trans-acting activity of OREB1 (Fig. 4d). Teo et al.[14] reported that a similar module regulates tuber induction in potatoes. It is plausible that regulatory modules consisting of these three protein family members contribute widely to various biological regulations.

By comparing the frequency of the *GF14h* haplotype (Fig. 5f and Supplementary Fig. 17b), we found that modern varieties tend to contain a higher frequency of LOF alleles (low germination rate) and old varieties contain a higher frequency of GOF alleles. The same tendency was more clearly observed for *qLTG3* (Fig. 5g and Supplementary Fig. 17c). These results suggested that LOFs of these two

genes might have been selected in the process of rice breeding for slow seed germination, and the selection pressure might be higher in northern areas of Japan and China. Then, why is slower seed germination preferred, especially in the northern areas? Rapid germinability has been a desirable trait as it reduces the risk of seedling mortality caused by temperature changes[40,41], which can be largely avoided with the development of temperature-controlled seed germination systems. However, rapid seed germination increases pre-harvest sprout risk[42,43]. Hence, breeders might replace the GOF alleles with LOF alleles for rice growing expansion even though rapid germination under lower temperature conditions is desirable for direct-seedling rice cultivation systems[41]. The negative effect of rapid germination on increasing pre-harvest sprout risk is lethal and non-negligible. Yamaguchi et al.[43] proposed a QTL pyramiding strategy using these GOF alleles and seed dormancy genes, such as *Sdr1* and *Sdr4*[44], to develop new strains with high germination rate without increasing pre-harvest sprout risk. Thus, advances in science will provide the answer to this old problem.

## Methods

### Plant material and genotyping
*Japonica* rice panels comprising 164 varieties were collected from various sites in Japan and grown in the Togo Field, Field Science Centre, Nagoya University (Supplementary Data 1). The *qLTG3* ORF from genomic DNA of the 164 varieties was amplified using PCR to determine its genotype. PCR products were detected on a 3% agarose gel stained with ethidium bromide and sequenced.

### Evaluation of germination rate
Seeds of each variety were collected 45 days after flowering, air-dried, stored at 45 °C for 3 days to break dormancy, and subsequently incubated in a growth chamber at 30 °C for 24 h or 15 °C for 96 h under dark conditions to induce germination. Germination was considered to have occurred when the epiblast was broken and the white embryo had emerged to a certain length (Supplementary Fig. 1). Germinated seeds were counted and the germination rates (%) were calculated by dividing them with those germinated at 30 °C for 48 h, under which all viable seeds were thought to have germinated.

### Sequencing and polymorphism calling
Additional genome information of the varieties used in this study (Supplementary Data 1) was retrieved as follows: genomic DNA of each accession was isolated from leaves using a DNeasy Plant Mini Kit (Qiagen, #69104) and fragmented into approximately 500 bp using Covaris S2 (Covaris). The NEBNext DNA Library Prep Reagent Set (BioLabs, #E6000) was used for DNA library construction. Paired-end sequencing was performed using the Illumina Hiseq system (Illumina Co., Ltd) with a read length of 100–150 bp. All reads were mapped against Os-Nipponbare-Reference-IRGSP-1.0 pseudomolecules (all.con ver.7, downloaded from http://rice.plantbiology.msu.edu/pub/data/Eukaryotic_Projects/o_sativa/annotation_dbs/pseudomolecules/version_7.0/all.dir/), and fastq files were converted into sam files using the bwa-mem command of BWA software ver0.7.18[45]. Commands samtools-view, samtools-sort, and samtools-index of Samtools software ver1.6[46] were used to generate, sort, and index bam files successively. The variants for each accession were called using the GATK HaplotypeCaller (release 4.0.4.0) with the '.g.vcf' extension[47]. GATK GenomicsDBImport and GenotypeGVCFs were used for joint genotyping to produce a single VCF per sample of GVCF. Heterozygous single nucleotide polymorphisms (SNPs) were first set to missing values before filtering for 5% as minor allele frequency, low mapping quality (<40), and 40% as the minimum count.

### Tri-allelic variants to tri-lines processing
Tri-allelic variants in a raw VCF file, which was the output of GATK GenotypeGVCFs, were transformed to three bi-allelic variants by our in-house Perl script after adding the information of the alleles at 220116 on Chr.3. The elements of the new lines other than Ref (Row 5), Alt (Row 6), and each genotype information on the accessions (from Row 10 onward) were duplicated from each original tri-allelic line. After this treatment, the heterozygous SNPs were transformed and filtered as described above.

### Heritability
The narrow-sense heritability ($\hat{h}^2$) of germination rate in each environmental condition was estimated using Eq. 1:

$$\hat{h}^2 = \hat{\sigma}_G^2 / \left( \hat{\sigma}_G^2 + \hat{\sigma}_\varepsilon^2 \right) \tag{1}$$

where $\hat{\sigma}_G^2$ and $\hat{\sigma}_\varepsilon^2$ are the genetic and error variances, respectively. These variance components were estimated by solving Eq. 2:

$$\boldsymbol{V} = \boldsymbol{G}\hat{\sigma}_G^2 + \boldsymbol{I}\hat{\sigma}_\varepsilon^2 \tag{2}$$

where $\boldsymbol{V}$ is the phenotypic variance; $\boldsymbol{I}$ is an identity matrix; $\boldsymbol{G}$ is the genetic relationship matrix calculated by function 'A.mat' in the R package rrBLUP version 4.3[47,48]. The solution of Eq. (2) was obtained by using function 'mixed.solve' in the R package rrBLUP version 4.3[49,50].

### GWAS
The GWAS for each environmental condition was performed using the function 'GWAS' in the R package rrBLUP version 4.3 with default parameter settings except for n.PC=5[49]. The GWAS for each environmental condition was performed based on the linear mixed model (LMM)[51]. Manhattan plots and quantile-quantile (Q-Q) plots with -log₁₀ *P*-values analysed by LMM were generated using the R package qqman[52]. This study's genotype-by-environment (G × E) GWAS was performed based on the recommended method in Yamamoto and Matsunaga[33]. The null hypothesis of the G × E GWAS in this study is that a marker has a common effect in both experimental conditions (i.e., 15 °C and 30 °C) but does not have an effect specific to each experimental condition. Therefore, the null model is as follows:

$$\boldsymbol{y} = \boldsymbol{Tt} + \boldsymbol{Ss} + \boldsymbol{x}\alpha + \boldsymbol{u}_G + \boldsymbol{u}_{GE} + \boldsymbol{\varepsilon} \tag{3}$$

where $\boldsymbol{y}$ and $\boldsymbol{\varepsilon}$ indicate $n \times 1$ vectors for phenotypic values and residuals, respectively; $n$ is the number of phenotypic records; $\boldsymbol{T}$ is an $n \times 2$ design matrix that assigns phenotypic values to the two experimental conditions, 15 °C and 30 °C, and $\boldsymbol{t}$ is a $2 \times 1$ vector of the population-wide mean for each experimental condition; $\boldsymbol{S}$ is an $n \times 5$ matrix whose column elements are the first five eigenvectors from principal component analysis of genotype data from all markers. $\boldsymbol{s}$ indicates a $5 \times 1$ vector of fixed effects for $\boldsymbol{S}$. $\boldsymbol{x}$ is an $n \times 1$ vector of genotype values of a marker coded as {−1, 1} = {REF/REF, ALT/ALT}; α is a marker effect common to both experimental conditions. $\boldsymbol{u}_G$ models the random effects common to both experimental conditions:

$$\boldsymbol{u}_G \sim MVN\left(\boldsymbol{0}, \left[\boldsymbol{Z}_G \boldsymbol{G} \boldsymbol{Z}_G'\right]\sigma_G^2\right) \tag{4}$$

where $MVN$ is the multivariate normal distribution; $\boldsymbol{Z}_G$ represents an $n \times m$ incidence matrix for the phenotype and random effects; $m$ is the number of varieties (i.e., $m = 164$ in this study); $\boldsymbol{G}$ is the $m \times m$ genetic relationship matrix calculated by function 'A.mat' in the R package rrBLUP version 4.3[48,49]; $\sigma_G^2$ is the variance for $\boldsymbol{u}_G$. $\boldsymbol{u}_{GE}$ models the G × E random effects as follows:

$$\boldsymbol{u}_{GE} \sim MVN\left(\boldsymbol{0}, \left[\boldsymbol{Z}_G \boldsymbol{G} \boldsymbol{Z}_G'\right] \circ \left[\boldsymbol{Z}_E \boldsymbol{Z}_E'\right]\sigma_{GE}^2\right) \tag{5}$$

where $\boldsymbol{Z}_E$ is the $n \times 2$ incidence matrix for the phenotypic values and environmental differences (i.e., 15 °C or 30 °C in this study); $\sigma_{GE}^2$ is the

variance for $u_{GE}$; the symbol ∘ indicates the Hadamard product for the left and right vectors or matrices. The alternative hypothesis of the G × E GWAS in this study is that a marker has different effects between the experimental conditions. Therefore, the alternative model is as follows:

$$y = Tt + Ss + \sum_{l}^{2}\{(\pi_l \circ x)\zeta_l\} + u_G + u_{GE} + \varepsilon \qquad (6)$$

where $\pi_l$ is an $n \times 1$ vector containing indicator variables that determines whether the phenotypic value is obtained from the $l$-th experimental conditions {1} or not {0}; and $\zeta_l$ is the marker effect in the $l$-th experimental conditions. The statistical significance of marker G × E effects was evaluated using the log-likelihood (LL) ratio test (LRT). The $P$-value of the G × E effect for each marker was calculated using the chi-square test based on the deviance D:

$$D = -2 \times (LL_{Eq.7} - LL_{Eq.4}) \qquad (7)$$

with the degree of freedom equal to 1. The theoretical validity and algorithms to estimate each parameter and fit the models are detailed in Yamamoto and Matsunaga[33].

### Linkage disequilibrium analysis
Linkage disequilibrium (LD) analysis using the R package LD heatmap[53] was used to define LD blocks surrounding significant peaks by confidence intervals. The VCF file after tri-allelic variants to tri-line processing was used for LD analysis.

### Candidate gene isolation
The position of polymorphisms in the rice genome was determined according to the information files of the rice gene locus (all.locus_brief_info.7.0) and genomic features (all.gff3) that were obtained from the rice genome annotation project (http://rice.plantbiology.msu.edu/pub/data/Eukaryotic_Projects/o_sativa/annotation_dbs/pseudomolecules/version_7.0/all.dir/). The genes annotated as "retrotransposon protein" or "hypothetical protein" were omitted.

### Phylogenetic analysis
Amino acid sequence alignments and estimation of phylogenetic topology were conducted as previously described in Yoshida et al.[54]. Bayesian Markov chain Monte Carlo (MCMC) analyses used flat priors and were run for 2,000,000 generations and four Markov chains (using default heating values) and were sampled every at 1000 generations. The initial 500,000 generations were discarded as burn-in.

### Plasmid construction
PCR amplification for all constructs was performed using high-fidelity PrimeStar DNA Polymerase (Takara). The primer sequences are listed in Supplementary Table 5. PCR-amplified fragments were sequenced to ensure that no mutations were introduced. Plasmids containing OREB1 mutants (OREB1 [S385A] and OREB1 [S385E]) were produced by site-directed mutagenesis[55] from plasmids containing the normal OREB1 sequence.

### Plasmids for GF14h complementation
To produce the GF14h complementation construct, a 7075 bp genomic DNA fragment containing the full-length GF14h gene was PCR-amplified from cv. Sensho genomic DNA containing Hap.2 of GF14h as a template. This fragment was cloned into pENTER/D-TOPO (Invitrogen). DNA fragments were then subcloned using the LR reaction into the Gateway binary vector pGWB502 without the 35 S promoter[56].

### Plasmids for transgenic experiments
For constitutive expression of GF14h, the coding sequences of GF14h were PCR amplified from the cDNA of Sensho, and PCR products were cloned into pENTER/D-TOPO. The GF14h cDNA fragments were subcloned using the LR reaction into the Gateway binary vector pUBQ-pGWB502. For knockout of MFT2, the synthetic genomic RNA (gRNA) inserted into the CRISPR/Cas9 plasmid was constructed according to Endo et al.[57] and transformed into NPB. The 20-nt oligonucleotides targeting the MFT2 sequences were annealed and cloned into the pU6gRNA-oligo vector using the two BbsI restriction sites. Next, a gRNA expression cassette in the pU6gRNA-oligo vector was ligated into a gRNA/Cas9-expressing binary vector (pZDgRNA_Cas9ver.2_HPT) using AscI/PacI sites. To confirm the introduction of the MFT2 mutation, genomic DNA was extracted from transformed regenerates, and the MFT2 locus was amplified using specific primers. PCR products were sequenced, and the on-target mutation on MFT2 was confirmed.

For knockout of MFT2, the synthetic genomic RNA (gRNA) inserted into the CRISPR/Cas9 plasmid was constructed according to Endo et al.[57] and transformed into NPB. The 20-nt oligonucleotides targeting the MFT2 sequences were annealed and cloned into the pU6gRNA-oligo vector using the two BbsI restriction sites. Next, a gRNA expression cassette in the pU6gRNA-oligo vector was ligated into a gRNA/Cas9-expressing binary vector (pZDgRNA_Cas9ver.2_HPT) using AscI/PacI sites.

### Plasmids for promoter-GUS experiment
For the pHap.2::GUS construct, the promoter region (2115 bp upstream from the OsGF14h start codon) was PCR-amplified from Sensho genomic DNA and cloned into pENTER/D-TOPO (Invitrogen). DNA fragments were then subcloned into the Gateway binary vector Δp35s-GUS/pGWB502 using Gateway LR Clonase II Enzyme mix (Invitrogen).

### Plasmids for Y2H assays
For the Y2H assay, pGADT7 (Clontech) and pGBKT7 (Clontech) were used as expression vectors. Hap.2 and Hap.1 of GF14h (GF14hs) were PCR-amplified using cDNA of Sensho and NPB, respectively, and cloned into the EcoRI-SmaI sites of pGBKT7. MFT2 CDS was amplified using cDNA of NPB and cloned into pGBKT7 in the same way.

For the construction of pGADT7-OREB1, OREB1 CDS with a SmaI and BamHI site at each end were cloned into the pGADT7 vector. TRAB1 was PCR-amplified using cDNA from NPB and cloned into the EcoRI-SmaI sites of pGADT7. GF14h and OREB1 were PCR-amplified using GF14h/pGBKT7, OREB1/pGADT7, and MFT2/pGBKT7 plasmids as templates; GF14h, OREB1, and MFT2 were cloned into pENTER/D-TOPO, respectively. DNA fragments were then subcloned into the Gateway binary vector pDEST22 as the AD vector and pDEST32 as the BD vector using Gateway LR Clonase Enzyme mix (Invitrogen). MFT2 CDS was amplified from MFT2/pGBKT7, and PCR products were cloned into pENTER/D-TOPO and then subcloned into the Gateway binary vector pYES-DEST52.

### Plasmids for subcellular localisation assay
The sGFP sequence was amplified and inserted into the SpeI-StuI site of pE2113_GW_SAS[58] to generate a vector for the protein fused with sGFP, pE2113_GW_sGFP. PCR-amplified DNA sequences of the genes were introduced into the XbaI-SpeI site of the vector. The mScarlet sequence was amplified and inserted into the SpeI-StuI site of pE2113_GW_SAS to generate a vector for expressing mScarlet fused protein, pE2113_GW_mScarlet. The CDS of OsPUP1 (Os03g0187800)[59], OsIDD2 (Os01g0195000)[60] and OsNST1 (Os02g0614100)[61] amplified with cDNA of NPB were introduced into the XbaI-SpeI site of the vector to produce the subcellular marker constructs for endoplasmic reticulum, nucleus, and golgi apparatus, respectively.

## Plasmids used for the BiFC experiment

PCR-amplified DNA sequences of GF14hs, OREB1, OREB1 mutants (OREB1(S385A, S385E)), and MFT2 were cloned individually into pENTER/D-TOPO vectors. Additionally, a 35S promoter sequence was spliced upstream of each gene using suitable restriction enzymes. Splicing was performed before or after the genes were cloned into the pENTER/D-TOPO vector, depending on the availability of sequenced pUC19 vector constructs. This involves digesting the source vector to release the DNA fragment for 35S, which is then subcloned into another pUC19 construct or directly spliced into a pENTER/D-TOPO vector construct. Once the pENTER/D-TOPO construct with the 35S sequence and the required gene without the STOP codon was ready, it was mixed with the Gateway vectors that possessed complementary fragments of YFP (nYFP and cYFP) with LR clonase II (Thermo Fisher). The pGWcY vector was used to express cYFP-fused proteins. GF14h/pnYGW and OREB1/pGWnY were prepared for the nYFP-fused proteins. MFT2/GWnY and MFT2/pnYGW were prepared for the nYFP-fused MFT2 to confirm their interaction with OREB1-cYFP and GF14h-cYFP, respectively[62].

## Plasmids for transient reporter assay

A fused OsEM motifA-CMV35s (core sequence) fragment in a pIG46 construct (gifted from Tsukaho Hattori, Nagoya University) was amplified by PCR using primers with HindIII and SmaI restriction sites. The resulting fragment was double-digested with the aforementioned enzymes and then spliced into a similarly digested pUC19 construct containing hRluc and nos terminator downstream of the insertion site. The resulting construct was validated by restriction enzyme digestion and DNA sequencing. For the effector plasmid, GF14hs and OREB1 CDS were cloned into the XbaI-SpeI site of pE2113_GW_SAS as previously described in Yoshida et al.[58] using NEBuilder HiFi DNA Assembly master mix (New England BioLabs). The coding sequence of MFT2 was amplified from the cDNA of NPB, and PCR products were cloned into the BamHI/SmaI site of pACT/pUC19[63] to produce pACTMFT2/pUC19.

## Plasmids for co-IP experiment

To express proteins with a desired tag protein at the N-terminus, the SpeI-sGFP (lacking the stop codons)-StuI-XbaI sequence was introduced into the XbaI-SacI site of pE2113_GW_SAS[58] using NEBuilder HiFi DNA Assembly master mix, resulting in a pE2113_n_sGFP vector. For the co-IP experiment, the coding sequence of 3xFLAG was amplified by PCR using 3xFLAG/pCAMBIA as a DNA template[64], and the amplified FLAG sequences were inserted into the SpeI-StuI site of pE2113_n_sGFP using NEBuilder HiFi DNA Assembly master mix to generate the pE2113_n_FLAG vector. The coding sequence of GF14h was PCR-amplified and cloned into the XbaI-SacI site of pE2113_n_FLAG. HA-GF14h and cMYC-MFT2 were first amplified by PCR using GF14h/pGBKT7 and MFT2/pGADT7 plasmids as templates, HA-GF14h or cMYC-MFT2 were cloned into the SpeI-SacI site of OREB1/pE2113_n_FLAG.

## Transgenic plants

Each construct was introduced into A. tumefaciens strain EHA105 and used to infect rice calli, according to Ozawa et al.[65]. These EHA105 stains were grown for 3 days on the AB medium containing 50 mg/L hygromycin and 50 mg/L kanamycin or 100 mg/L spectinomycin solidified with 1.5% agar. The bacterial cells were resuspended in AAM medium. The calli of NPB were soaked in this suspension for 2 min and blotted dry, using sterile Kimwipes. Then these calli were transferred on filter paper placed on a co-culture medium which was prepared by spreading 5.35 mL of a liquid 2N6-AS medium on a bottom medium. After co-cultivation, the calli were washed three times with water containing 0.6 mg/L acetosyringone. After washing, the transgenic calli were selected on a medium containing 25 mg/L meropenem

and hygromycin. Seedlings were established on Murashige and Skoog (MS) medium [half-strength MS salt mixture (pH 5.7), B5 vitamins, 1% sucrose, and 0.8% gellan gum]. Transgenic seedlings were grown to maturity in pots under greenhouse conditions at 28 °C and 60% humidity under 16 h or 10 h light conditions.

## GUS staining

GUS staining of the transgenic plants carrying pGF14h^Hap.2::GUS was performed as described by Morii et al.[66]. The tissues of transgenic plants were stained by vacuum infiltration for 10 min with a GUS staining buffer [50 mM sodium phosphate, pH 7.0, 1 mM 5-bromo-4-chloro-3-indolyl-β-D-glucuronide and 7% (v/v) methanol]. After incubation in darkness for 18 h at 37 °C, the seedling samples were completely cleared with 70% ethanol. The staining was observed under a microscope (Olympus SZX-12).

## Germination test

Seeds of transgenic plants were spread onto plates containing 20 mL of distilled water or ABA solution. The plates were placed in a chamber at 15 °C or 30 °C. Germination was defined as the emergence of the radical, and the number of germinated seeds was counted every 24 h. The germination rate was calculated as the number of total germinated seeds at the time point divided by the number of total seed (20-30 seeds).

## Expression analysis by quantitative RT-PCR

Total RNA was isolated from seeds at 12 h after imbibition using an RNeasy Plant minikit (74904, Qiagen). The first strand of cDNA was synthesized from 0.5 μg of total RNA using an Omniscript reverse transcription kit (205113, Qiagen). Transcripts were quantified by quantitative RT-PCR (qRT-PCR) analyses using one-twentieth of the resulting cDNA as template. qRT-PCR was performed with a CFX96 real-time PCR detection system (Bio-Rad) with the SYBR Green PCR kit (1725150, Bio-Rad) and appropriate primers (Supplementary Table 5). Relative transcript abundance was calculated by CFX96 manager software (Bio-Rad).

## Yeast-two-hybrid assay

A yeast-two-hybrid (Y2H) assay was performed according to Ueguchi-Tanaka et al.[67] using the BD Matchmarker Two-Hybrid System 3 (Clontech; Yeast Protocols Handbook #PT3024-1). Vector cassettes for DNA-BD and -AD (activation domain) were used as negative controls, and Saccharomyces cerevisiae strain Y187 or PJ69-4A was used as the host. Selection medium lacking histidine was used for Y2H and Y3H. Experiments were independently repeated at least three times. β-gal activity was determined using a liquid assay with yeast (Y187) transformants.

## Subcellular localisation assay

To detect the subcellular localisation of GF14hs and GF14h^Hap.2-MFT2, the above-mentioned marker constructs were purified and then introduced into rice mesophyll protoplasts with GFP-GF14hs or nYFP-GF14h and cYFP-MFT2 constructs according to the previous study[63]. Plasmids were extracted using NucleoBond Xtra Midi according to the manufacturer's instructions (Takara, U0412). Dehulled seeds of NPB were sterilized with 75% ethanol for 1 min. These seeds were further sterilized with 2.5% sodium hypochlorite for 20 min, washed at three times with sterile water and then grown on half-strength MS medium with a photoperiod of 12 h light (about 150 μmol m$^{-2}$ s$^{-1}$) and 12 h dark at 26 °C for 6–7 days. Green tissues from the leaf sheath of 80-100 rice seedlings were used. The leaf strips which were cut using razors were immediately transferred into 0.6 M mannitol for 10 min in the dark. After the solution, the strips were incubated in an enzyme solution (1.5% Cellulase RS, 0.75% Macerozyme R-10, 0.6 M mannitol, 10 mM MES at pH 5.7, 10 mM CaCl$_2$ and 0.1% BSA) for 4 h in the dark with

gentle shaking (80 rpm). After the enzymatic digestion, an equal volume of W5 solution (154 mM NaCl, 125 mM CaCl$_2$, 5 mM KCl and 2 mM MES at pH 5.7) was added. Protoplasts were released by filtering through 40 µm nylon meshes with 3–5 washes of the strips using W5 solution. The pellets were collected by centrifugation at $500 \times g$ for 3 min. After washing once with W5 solution, the cells were then resuspended in MMG solution (0.4 M mannitol, 15 mM MgCl$_2$ and 4 mM MES at pH 5.7) at a concentration of $2 \times 10^6$ cells mL$^{-1}$. For each sample, 5–10 µg of plasmids were mixed with 100 µL MMG solution with protoplasts (about $2 \times 10^5$ cells). 110 µL PEG solution [40% (W/V) PEG 4000, 0.2 M mannitol and 0.1 M CaCl$_2$] were added, and the mixture was incubated at room temperature for 20 min in the dark. After incubation, 440 µL W5 solution were added slowly. The resulting solution was mixed well by gently inverting the tube, and the protoplasts were pelleted by centrifugation at $500 \times g$ for 3 min. The protoplasts were resuspended in 1 mL WI solution (0.5 M mannitol, 20 mM KCl and 4 mM MES at pH 5.7). For analysis of localisation at plasma membrane, the transfected protoplasts were treated with FM4-64 (T13320, ThermoFisher) according to the manufacture instructions. The protoplasts were incubated for 24 h at 22 °C after transfection. Fluorescence signals were captured using a Zeiss LSM 700 laser microscope with ZEN 2.3 software.

### BiFC assay
A BiFC assay was performed using protoplasts prepared by the same procedure as the subcellular localisation assay mentioned above[58,68]. The protoplasts were incubated for 24 h at 22 °C after transfection unless otherwise mentioned. Successful complementation was detected using a Zeiss LSM 700 laser microscope using laser light with a wavelength of 488–555 nm.

### Transient reporter assay
Transient reporter assays using protoplasts prepared as the subcellular localisation assay mentioned above. The fLUC and hRLUC activities were measured using the Dual-Luciferase Reporter Assay system (E1910; Promega) and Luminoskan Ascent (Thermo Scientific) according to the manufacturer's protocol. The relative ratio was determined by comparing this ratio with that obtained with the empty vector. The mean relative ratios were calculated based on the data derived from three independent experiments.

### Co-immunoprecipitation assay
Co-immunoprecipitation (Co-IP) was performed using protoplasts prepared by the same procedure as the subcellular localisation assay mentioned above and the previous studies[63,69,70]. Isolated mesophyll protoplasts ($2 \times 10^6$) were transfected with 15 µg of DNA plasmid mixture (5 µg of plasmids) and incubated overnight. Total proteins were extracted from the protoplasts by sonicating in IP buffer (25 mM Tris-HCl pH 8, 1 mM EDTA, 50 mM NaCl, 5% Triton X-100, and 1× protease inhibitor). After centrifugation at $20,000 \times g$ for 10 min, 100 µL of supernatant was mixed with 2 µL of anti-HA (Sigma; H3663) and incubated for 1 h at 4 °C. Then, 10 µL of Dynabeads protein G solution (Thermo Fisher Scientific; DB10003) was added and incubated again for 1 h at 4 °C. The beads were then washed five times with IP buffer without Triton X-100, and samples were analysed by immunoblot using anti-FLAG M2-HRP (1:5000; Sigma; A8592) and anti-HA-HRP antibodies (1:4000; Sigma; H6533).

### Haplotype network analysis
Haplotypes were defined as the groups encoding unique amino acid sequences in more than two varieties. Haplotype network analysis was performed using PopART software based on the 'Minimum Spanning Network' approach[71]. The origins of Hap.7 and Hap.8 were manually inspected from the genome region around the GF14h gene.

### Reporting summary
Further information on research design is available in the Nature Research Reporting Summary linked to this article.

### Data availability
The sequence data has been deposited in DNA Data Bank of Japan Sequence Read Archive (DRA) under accession DRA012529, DRA004358, DRA008452, and SRA1155070. The previously published 3000 *O. sativa* genome raw sequencing data are available from GigaScience Database [https://doi.org/10.5524/200001]. The previously published *O. rufipogon* genome raw sequencing data are available at OryzaGenome [http://viewer.shigen.info/oryzagenome2detail/downloads/index.xhtml], RiceHap3 and RicePanGenome database [http://www.ncgr.ac.cn/RicePanGenome]. Source data are provided with this paper.

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

## Acknowledgements

We thank Miyako Ueguchi-Tanaka (Nagoya University) for helpful advice. This work was supported by Grants-in-Aid for Scientific Research on Innovative Areas (16H06468 to M. Makoto) and KAKEN (17J09723 and 21K15120 to H.Y; 22H02294 to M. Makoto.) from Japan Society for the Promotion of Science and by Cabinet Office, Government of Japan, Moonshot Research and Development Program for Agriculture, Forestry and Fisheries (funding agency: Bio-oriented Technology Research Advancement Institution) (JPJ009237 to H. Y, E. Y., and M. Matsuoka).

## Author contributions

H.Y., K.H., K.Y., E.Y., and M. Matsuoka designed the study. H.Y., K.H., K.Y., M. Mori, and E.Y. carried out the GWAS analyses. H.Y., K.H., M.K., and E.K. examined the seed germination and performed the transgenic experiments. H.Y., K.H., K.Y., M.K., M.H., R.L.O., and P.H. elucidated the molecular mechanisms. H.Y., K.H., and M. Mori performed the population genetic analyses. H.Y., K.H., F.W., M.K., R.L.O., E.Y., and M. Matsuoka wrote the manuscript.

## Competing interests

The authors declare no competing interests.
