## [Peer Review File · Nature Communications]

Genome-wide association study identifies a gene responsible for temperature-dependent rice germinationReviewers' Comments:

Reviewer #1:

Remarks to the Author:

A G×E GWAS analysis detected a contribution of GF14h on chromosome 11 under high temperature cultivation (30C), and further investigation found that a 4bp InDel in the coding region of the gene caused a change of the germination rate depending on temperature. This gene made a complex with OREB1 and MFT2 to control germination rate by regulating ABA-responsive genes. The authors also claimed that this gene contributes to the geographical adaptation of rice.

I found that this study is statistically sound, the findings are presented clearly, the discussion makes sense, and the conclusion is rational. The only thing I would suggest is to expand the scope of the introduction. Currently, it is too short and needs more information.

Below are some minor issues:

- It seems that Fig. 1d (after the conversion of variants) is the same as Fig. 1a—please double-check.
- Extended Fig.3: The Q-Q plots do not look good to me—suitable data transformation may necessary prior to the GWAS analysis to improve the power of the analysis.
- P4, line 76: Change "Extended Data Fig. 4d-f" to "Extended Data Fig. 4."
- P4, lines 79-80: How big is the peak 1 region?
- Fig. 2: "...indicate the candidate region for Peak 1,..." change to "...indicate the close up of Peak 1 region,..."
- Extended Table 4: Please add in the footnotes what is "." referring to in the first column of Hap3 and Hap4. Is it the 71 bp deletion? Also, among those accessions (especially the japonica and indica), how many are cultivated, landraces and/or weedy?
- Fig. 3a and b: I assume that these experiments were performed at 30C?

Reviewer #2:

Remarks to the Author:

Several previous reports have explored the candidate loci related to cold and heat stress tolerance in rice using GWAS. However, the underlying regulatory mechanism remains poorly understood. Yoshida et al., conducted a G×E GWAS with the germination rate of Japanese rice cultivars under high and low temperature conditions. The authors found that a 4 bp-InDel in one 14-3-3 protein (GF14h) was associated with the germination rate of rice under high temperature. They further showed that GF14h constitutes a transcriptional regulatory module with OREB1 and MLF2 to control the germination rate by regulating ABA responsive genes. Overall, the GWAS strategy provided impressive data; however, the molecular basis of GF14h-OREB1-MFT2 module in regulating germination under high temperature conditions is still not clear, and some conclusions could not be fully supported by current evidence. Specific comments are listed below.

1. In Fig 2c, the authors showed that germination rates of GF14h Hap.1 are lower than Hap.2/Hap.3 plants at 30C. If GF14h and OREB1 module was involved in this process, whether the ABA response is different between GF14h Hap.1 and Hap.2/3?

2. What's the germination phenotype of OREB1 and MFT2 loss-of-function mutants at high temperature? Given that GF14h work together with OREB1 and MLF2 to control the temperature-dependent seed germination rate by regulating ABA responsive genes, the authors should test whether disrupt GF14h, OREB1 or MLF2 affect ABA response and ABA-responsive gene expression under high temperature.

3. Another major concern is that this study lacks genetic evidence to establish the interaction of GF14h, OREB1 and MFT2.

4. In Fig 3c, the author showed that GF14h Hap.2 but not Hap.1 could interact with OREB1. This observation is interesting, but more in vivo evidence should be provided to support the conclusion. The effect of high temperature on their interaction should be examined. The quality of BiFC assays needs to be improved. The subcellular compartments marker or dye should be used.

5. In Fig 4b, the authors demonstrated that the BiFC signal of GF14h-MFT2 was "dot-like" in the

cytosol. I don't think it is a professional description about these "dot-like" speckles. The authors should figure out what subcellular compartment they are?

6. In Fig 4d, the authors showed that the transcriptional activity of OREB1 was partially diminished by GF14h in Y2H system. I am wondering whether the protein stability or the DNA binding affinity of OREB1 is affected by GF14h. Based on this observation, more biochemical experiments are necessary to explore the biological significance of GF14h-OREB1 interaction.

7. It seems that MFT2 could release the suppressive effect of GF14h of OREB1 in Fig 4d; however, GF14h-OREB1 interaction and subcellular localization was not affected by the presence of MFT2 (Fig 4f and 4h). How to explain their antagonistic relationship? And how high temperature affects this suppression?

8. The model could be improved. It should add the connection of GF14h-OREB1-MFT2 module with both temperatures and rice natural variations.

9. The description of G×E GWAS method is confusing. For example, X is only the design matrix of SNP, but β represents the SNP effect and population structure, these two descriptions seem inconsistent. The author mentioned that "g models the genetic background...", it should be "Z" in the formula, but the relationship between "g" and "Z" is unclear. Also, the signs are confusing, what's the meaning of "g", " β g", and " β G"?

10. In addition to G×E GWAS results, the original GWAS results should also be shown.

11. In Extended Data Fig. 2e, the authors claimed that population structure does not have much influence on the data. This would require a more statistical approach as PCA is a visualization approach.

Reviewer #3:

Remarks to the Author:

Using genome-wide association studies and molecular biology approaches, the authors explored genes controlling seed germination rates in a temperature-specific manner. The primary candidate is GF14h, a known germination regulator in barley, and the authors suggested that lines carrying the loss of function allele have slower germination under higher temperatures. Seed germination is clearly a significant trait for local adaptation, and understanding the genetic basis is essential, especially for crop breeding and revealing the domestication process.

Major comments

Molecular biology experiments using transgenic lines support the involvement of GF14h in seed germination. However, statistics and population genetics analyses are quite insufficient, and a lot of works are needed to prove the g x e effects.

Genome-wide association study

The authors performed three GWAS tests, including g x e, 30°C, and 15°C. According to Extended data, Fig. 3, g x e and 15°C tests show overinflation of p-values that means failed model fitting. For the linear mixed models like the authors used, the transformation of phenotypes into the most normal (i.e., box-cox) is necessary to reduce false-positive errors from the population structures. The unfitted g x e test seems to be affected by the skewed distribution of 15°C phenotypes.

g x e model

Although the GWAS result is not solid, both 30°C and 15°C (Fig 1b and c) show the same peak around GF14h. It means the peak has common G effects on seed germination rate under both temperatures unless the direction is opposite. Why is the temperature-specific G effect (G x E) so strong in Fig 1a? I think because the common g effect is missing in this G x E model. Usually, Y is defined as the sum of G, E, G x E, Z and error (Z is kinship). Thus, the full model is written as $Y = u + Xb_g + Xb_g \times e + Z + e$, and the null model to test $b_g \times e$ is $Y = u + Xb_g + Z + e$. If I understand the model in this study (L426 - 432) correctly, the authors used $Y = u + Xb_g \times e + Z + e$, and the null model is $Y = u + Z + e$, as they wrote 'G x E is included in X, L429'. In this case, G x E is overestimated because it

partially contains common g effects as well.

Ref. i.e.,

Korte et al., Nature genetics (2012)

<https://www.nature.com/articles/ng.2376>

Moore et al., Nature genetics (2019)

<https://www.nature.com/articles/s41588-018-0271-0>

Identification of the causality

The peak1 is very wide, over 500kbp. How many genes are involved in the region? Why can authors exclude possibilities that causality affects traits through gene expression regulation? Please clarify whether the authors claim GF14h is only a causal gene in the haplotypes or not. Also, how much phenotypic variation is explained by the peak1 and the 4bp deletion in GF14h?

Fig 2c-d, The raw values contain effects of population structures. Please use BULP to show the allelic effects.

L217, Are the authors suggesting allelic heterogeneity? Instead of "the novel approach", the traditional co-factor model works in the same way and that's more straightforward for the case.

<https://www.nature.com/articles/nature08800>

Actually, analyses using synthesized SNPs based on the functionality are not a novel approach.

What is the reason that the low germination rate of transgenic lines with Hap2 (Fig2e and f) compared with the natural lines (Fig2c and d) at 24h? In addition, if you compare germination rate at 30°C 24h and 15°C 96h, Hap2 genome/NBP might have the same effect in both temperatures.

L266, How did the authors estimate the selection pressure? EHH does not take into account the population structure and the sampling bias. If the authors claim the allele is under positive selection, please show and compare the genome-wide significance. Without any population genetic analyses, the selection pressure is speculation (L266).

How is heritability (= genetic effect) under 15°C and 30°C?

Please explain the method to predict the haplotype in materials and methods.

Why can the authors conclude that GF14h in the Hap1 is the loss-of-function gene? There are many cases that the indels do not retrieve the gene function completely.

Reviewer #4:

Remarks to the Author:

In the manuscript entitled "Genome-wide association study identifies a gene responsible for temperature-dependent rice germination", the authors performed a genome wide association study to unveil the relationships between genotype and environment on the germination rate of rice cultivars. This work led to the identification of a gene encoding to the 14-3-3 isoform GF14h as a factor controlling germination in G x E dependent manner. Furthermore, they demonstrated that GF14h is a negative regulator of ABA signalling and that it can bind to a ABA-regulated transcriptional complex. Finally, the authors demonstrated that a GF14h loss of function allele contributes to the geographical adaptation of Japanese and chines cultivars.

1. I have found that the genetic studies leading to the identification of GF14h alleles (Figs 1 and 2) are well performed and convincing, as well as the experiment demonstrating that GF14h is a negative

regulator of ABA signalling. However, in my opinion a fundamental point is lacking: there is no functional demonstration that the function of 14-3-3 in the regulation of germination, emerged from genome-wide association studies, is carried out through the regulation of the transcriptional complex OREB1 / MFT2. In this way, the characterization of the interaction between GF14h, OREB1 and MFT2, although interesting, does not make a real contribution to provide a rationale for the identification of GF14h as an ABA-dependent regulator of seed germination. For example, in principle, GF14h could negatively regulate ABA signalling by activating a physiological process antagonistic to ABA (e.g., 14-3-3 have a role in the regulation of signalling and biosynthesis of GA, which play an opposite role respect to ABA in the germination process). I think that the authors should address this point or at least they should critically discuss in the manuscript.

2. Did the authors verify whether GF14Hap1 is normally expressed in the plant? The authors should investigate this aspect, in order to understand if the Hap1 allele is loss of function due to a lack of / insufficient expression or because the amino acid substitutions result in a lower ability of GF14Hap1 to interact with target proteins. In the Yeast Two-hybrid Assay, is the absence of β -galactodidase activity due to the lack of expression of the protein in yeast or rather to the lack of ability of GF14 Hap1 to interact with OREB1 and TRAB1? Again, in the protein-protein interaction experiments (e.g. BiFC and Co-IP experiments) did the authors test the properties of GF14 Hap1? The authors should address these points.

3. In the confocal microscope experiments, the interpretation of the localization of 14-3-3 is not entirely convincing (Fig. 3d, first lane). The nuclear signal, if present, is masked by the cytosolic one. Furthermore, it could also be present a signal at the plasma membrane level. This signal could be relevant for the purposes of the manuscript, as a known function of the 14-3-3 proteins is the regulation of the plasma membrane H⁺-ATPase, which plays an important role in the regulation of seed germination. I strongly suggest to use fluorescent markers for the nucleus and plasma membrane. Minor point: the reference of the scale bar is missing.

4. The introduction completely lacks a description of the molecular mechanism of 14-3-3 action. This makes it difficult to understand some experiments, e. g. Figs 3 and 4, where a reader non-expert in the 14-3-3 field may not understand that the S358A mutation prevents phosphorylation and consequently 14-3-3 association to a consensus motif, while the S385E mutation mimics phosphorylation. I strongly suggest to include in the introduction a short introduction on 14-3-3 including the essential bibliography, also totally missing.

Responses to Reviewer #1 Comments:

I found that this study is statistically sound, the findings are presented clearly, the discussion makes sense, and the conclusion is rational. The only thing I would suggest is to expand the scope of the introduction. Currently, it is too short and needs more information.

Thank you very much for your valuable suggestions. According to your comments, we have expanded the scope of introduction including current knowledge about 14-3-3 (P3, L46-68).

- It seems that Fig. 1d (after the conversion of variants) is the same as Fig. 1a—please double-check.

Thank you for the comment. The results presented in Fig. 1a and d was completely different.

The reason why both the results look similar may be because the number of polymorphisms did not change much before and after variant conversion. However, in the revised version, the GxE GWAS was re-performed and Fig. 1a and d were revised as pointed out by another reviewer.

- Extended Fig.3: The Q-Q plots do not look good to me—suitable data transformation may necessary prior to the GWAS analysis to improve the power of the analysis.

Thank you for your suggestion. We received a similar comment by Reviewer #3, so we re-ran the GWAS for germination rate at 15 °C after performing a box-cox transformation for making normal distribution (Supplementary Fig. 2c) and using the principal components as covariates. As a result, the Q-Q plot was greatly improved (Supplementary Fig. 3c, f).

- P4, line 76: Change “Extended Data Fig. 4d-f” to “Extended Data Fig. 4.”

Thank you for your suggestion. We revised our text accordingly (P5, L96).

- P4, lines 79-80: How big is the peak 1 region?

The size of the peak 1 is about 500 kbp. Fig. 2a shows the close-up view of the peak.

- Fig. 2: “...indicate the candidate region for Peak 1,...” change to “...indicate the close up of Peak 1 region,...”

Thank you for your suggestion. We revised our text accordingly (P35, L811-812).

- Extended Table 4: Please add in the footnotes what is “.” referring to in the first column of Hap3 and Hap4. Is it the 71 bp deletion? Also, among those accessions (especially the japonica and indica), how many are cultivated, landraces and/or weedy?

Thank you for the suggestion. “.” indicates the REF (Nipponbare)-type polymorphism. In the revised version, we replaced “.” with “REF” in Supplementary Table 4. As mentioned, this analysis was performed using the 3K panel reported by Wang et al., Nature (2018). This panel includes both landrace and cultivated (but not weedy), but unfortunately, we cannot provide the complete answer because the detail information is not tied to all the lines.

- Fig. 3a and b: I assume that these experiments were performed at 30C?

Yes, we performed these experiments at 30 °C. This information has been added in its legend.

Responses to Reviewer #2 Comments:

1. In Fig 2c, the authors showed that germination rates of GF14h Hap.1 are lower than Hap.2/Hap.3 plants at 30°C. If GF14h and OREB1 module was involved in this process, whether the ABA response is different between GF14h Hap.1 and Hap.2/3?

Thank you for the comment. We compared the ABA response of NPB, which contains GF14h^{Hap.1}, and NPB transformed with GF14h^{Hap.2} during seed germination. The results demonstrated that the transformation of GF14h^{Hap.2} into NPB dramatically reduced the ABA responsiveness (Fig. 3a), and the expression of three ABA responding genes, *OsRab16A*, *OsLea3* and *OsEM* (Fig. 3b). According to these results, there is substantial difference in the GF14h function in terms of ABA response between GF14h^{Hap.1} and GF14h^{Hap.2}. Furthermore, we conducted the additional experiments demonstrating that GF14h^{Hap.1} did not interact with OREB1 or MFT2 *in vivo*, and did not affect the transcriptional activity of OREB1 (Supplementary Fig. 10e and 15b). From the results above, we concluded that GF14h^{Hap.1} is a loss-of-function haplotype.

2. What's the germination phenotype of OREB1 and MFT2 loss-of-function mutants at high temperature? Given that GF14h work together with OREB1 and MFT2 to control the temperature-dependent seed germination rate by regulating ABA responsive genes, the authors should test whether disrupt GF14h, OREB1 or MFT2 affect ABA response and ABA-responsive gene expression under high temperature.

Thank you for the important comment. The effect of GF14h on ABA responsiveness at 30 °C has already been described above. The relationship between OREB1 and ABA response was revealed as follows: OREB1 (also named as OsABI5) is an ABA-dependent negative regulator of stress tolerance in rice (Skubacz et al., Front. Plant Sci., 2016); its homolog ABI5 in *Arabidopsis thaliana* is well known as a key regulator ABA-dependent regulation of seed germination (Skubacz et al., Front. Plant Sci., 2016); more recently, Li et al. (Frontiers in Plant Science, 2021) reported decreased germination rate of the

oreb1/abi5 mutant and decreased expression of ABA catabolic gene at 30 °C (As this paper was published after our submission, we added the sentences about the study in the revised manuscript). A recent study reported that *mft2* mutant exhibited delayed seed germination at 30 °C and reduced expression of the ABA responsive genes (Song et al., the Plant journal, 2020) and this has been mentioned in the previous our manuscript (P12, L279-281).

3. Another major concern is that this study lacks genetic evidence to establish the interaction of GF14h, OREB1 and MFT2.

We thought that the best way to investigate the genetic interaction of GF14h, OREB1, and MFT2 would be to combine each of these genes to create a mutant, but it was difficult to do so within the time for revision. Therefore, to investigate the effects of these genes on rice germination, we tried to adopt the method of transient particle bombardment, which Nakamura et al. had previously used to investigate the function of MFTs in wheat (Plant Cell, 2011). This method uses immature embryos, but we could not obtain a sufficient number of immature embryos in this winter season. Therefore, we attempted to establish a revised method using rice mature embryos detached from mature seeds. As a result, we succeeded in detecting luciferase activity in detached embryos transfected with the 35S::hrLUC plasmid, however, we could not establish conditions under which the activity was stably observed, perhaps because of the difficulty in stable detachment of the embryos. However, in some cases, high luciferase activity was observed (left figure below), so we used this system to examine the effects of OREB1 and GF14h on germination. The results showed that introduction of OREB1 into NPB embryos delayed germination, and simultaneous introduction of functional GF14h tended to suppress this effect of OREB1 (right figure below), supporting our germination model by GF14h and OREB1. As these results are preliminary by using an unstable system, we refrain from presenting them in this paper. However, all the results presented in this paper demonstrate that the interaction of GF14h, OREB1, and MFT2 to control the ABA-regulating gene expression. We also agree that the genetic interaction between GF14h, OREB1, and MFT2 pointed out by the reviewer is important for making our model more convincing, and we will confirm this by further experiments in near future.

4. In Fig 3c, the author showed that GF14h Hap.2 but not Hap.1 could interact with OREB1. This observation is interesting, but more *in vivo* evidence should be provided to support the conclusion. The effect of high temperature on their interaction should be examined. The quality of BiFC assays needs to be improved. The subcellular compartments marker or dye should be used.

Thank you for your important remarks. We performed an additional BiFC assay and found no interaction between GF14h^{Hap.1} and OREB1 or MFT2 (Supplementary Fig. 10e). To analyze the effect of temperature on the interaction of GF14h^{Hap.2} and OREB1, we performed BiFC assays incubated at 15 °C and 26 °C instead of 22 °C, because protoplasts destabilized over 26 °C temperature. However, the results showed that temperature did not substantially control these physical interactions (Supplementary Fig. 10f, g).

Followed by your suggestion, we also conducted the BiFC assays with the subcellular markers, which revealed that the complex of GF14h^{Hap.2} and MFT2 are mainly localized in ER but not in nucleus, golgi apparatus, and plasma membrane (Supplementary Fig. 10a-d).

5. In Fig 4b, the authors demonstrated that the BiFC signal of GF14h–MFT2 was “dot-like” in the cytosol. I don’t think it is a professional description about these “dot-like” speckles. The authors should figure out what subcellular compartment they are?

As mentioned in our response to the above suggestion, we confirmed that the complex of GF14h^{Hap2} and MFT2 are mainly localized in ER but not in nucleus, golgi apparatus, and plasma membrane (Supplementary Fig. 10a-d).

6. In Fig 4d, the authors showed that the transcriptional activity of OREB1 was partially diminished by GF14h in Y2H system. I am wondering whether the protein stability or the DNA binding affinity of OREB1 is affected by GF14h. Based on this observation, more biochemical experiments are necessary to explore the biological significance of GF14h-OREB1 interaction.

According to your suggestion, we examined whether GF14h alters the stability of the OREB1 in rice protoplasts. The increase of the GF14h^{Hap.2} level did not alter the amount of OREB1 (Supplementary Fig. 15e-g), demonstrating that GF14h^{Hap.2}-OREB1 interaction does not affect its stability.

7. It seems that MFT2 could release the suppressive effect of GF14h of OREB1 in Fig 4d; however, GF14h-OREB1 interaction and subcellular localization was not affected by the presence of MFT2 (Fig 4f and 4h). how to explain their antagonistic relationship? And how high temperature affects this suppression?

Thank you for your comment. We consider that GF14h functions as a co-repressor by interacting with OREB1, whereas MFT2 relieves the repressive function of GF14h. We predicted two hypotheses for the mechanism by which MFT2 releases the inhibitory function of GF14h. One possible explanation would be to prevent GF14h from recruiting other co-repressor(s) (such as TOPLESS), or to recruit additional transcriptional activation factor(s).

In relation to the effects of high temperature on these inhibitory mechanisms, recently Li et al. (2021) reported that ABA accumulates in large amounts at low temperatures. Such high levels of ABA promote the accumulation of OREB1 and mask the transcriptional effects of GF14h function. On the other hand, at high temperatures, low levels of ABA and OREB1 would allow GF14h to influence germination. The manuscript and Fig. 4i

have been revised to clarify these explanations (P11, L250-254).

8.The model could be improved. It should add the connection of GF14h-OREB1-MFT2 module with both temperatures and rice natural variations.

Thank you for the comment. As mentioned above, we revised the model as shown in Fig. 4i to explain our findings and hypothesis more clearly.

9.The description of G×E GWAS method is confusing. For example, X is only the design matrix of SNP, but β represents the SNP effect and population structure, these two descriptions seem inconsistent. The author mentioned that “g models the genetic background...”, it should be “Z” in the formula, but the relationship between “g” and “Z” is unclear. Also, the signs are confusing, what’s the meaning of “g”, “□g”, and “□G”?

Thank you for your suggestion. We have completely rewritten the method for the GxE GWAS, including the issues you pointed out as follows:

“The GWAS for each environmental condition was performed based on the following linear mixed model⁴⁹:

$$\mathbf{y} = \mathbf{S}\mathbf{s} + \mathbf{x}\alpha + \mathbf{u}_G + \boldsymbol{\varepsilon} \quad (3)$$

where, \mathbf{y} and $\boldsymbol{\varepsilon}$ indicate $n \times 1$ vectors for phenotypic values and residuals, respectively; \mathbf{S} is an $n \times c$ matrix whose column elements are the eigen vectors from principal component analysis of genotype data from all markers. In this study, eigen vectors from the first five principal components were used. \mathbf{s} indicates $c \times 1$ vector of fixed effects for \mathbf{S} . \mathbf{x} is an $n \times 1$ vector of marker genotype values coded as $\{-1, 1\} = \{AA, aa\}$; α is the marker effect. \mathbf{u}_G models the random effects as follows:

$$\mathbf{u}_G \sim MVN(\mathbf{0}, [\mathbf{Z}_G \mathbf{G} \mathbf{Z}'_G] \sigma_G^2) \quad (4)$$

where MVN is the multivariate normal distribution; and \mathbf{Z}_G represents an $n \times m$ incidence matrix for the phenotype and random effects. The genotype-by-environment ($G \times E$) GWAS was performed based on the recommended method in Yamamoto and Matsunaga (2021)³¹.

$$\mathbf{y} = \mathbf{T}\mathbf{t} + \mathbf{S}\mathbf{s} + \sum_l^L \{(\boldsymbol{\pi}_l \circ \mathbf{x})\zeta_l\} + \mathbf{u}_G + \mathbf{u}_{GE} + \boldsymbol{\varepsilon} \quad (5)$$

where \mathbf{z}_l is an $n \times 1$ vector containing indicator variables that determine whether the phenotypic value is obtained from l -th environment $\{1\}$ or not $\{0\}$. ζ_l is the marker effect in the l -th environment. \mathbf{u}_{GE} models the G×E random effects as follows:

$$\mathbf{u}_{GE} \sim MVN(\mathbf{0}, [\mathbf{Z}_G \mathbf{G} \mathbf{Z}'_G] \circ [\mathbf{Z}_E \mathbf{Z}'_E] \sigma_{GE}^2) \quad (6)$$

where \mathbf{Z}_E is the incidence matrix for the phenotypic values and environmental differences (i.e., 15 °C or 30 °C in this study); and σ_{GE}^2 is the variance for \mathbf{u}_{GE} . The model without marker G × E effect is as follows:

$$\mathbf{y} = \mathbf{Tt} + \mathbf{Ss} + \mathbf{x}\alpha + \mathbf{u}_G + \mathbf{u}_{GE} + \boldsymbol{\varepsilon} \quad (7)$$

The statistical significance of marker G×E effects was evaluated using the log-likelihood (LL) ratio test (LRT). The P -value of G × E effect for each marker was calculated using the chi-square test based on the deviance D :

$$D = -2 \times (\text{LL}_{\text{Eq.5}} - \text{LL}_{\text{Eq.7}}) \quad (8)$$

with the degree of freedom equal to $L-1$ (i.e., 1 in this study). The algorithms to estimate each parameter and to fit the models are given in detail in Yamamoto and Matsunaga (2021)³¹. Manhattan plots and quantile-quantile (Q-Q) plots with $-\log_{10} P$ -values analysed by LMM were generated using the R package qqman⁵⁰. Germination rate for 15 °C were used after Box-Cox transformation optimized in terms of normality. For G × E GWAS, the germination rate at each temperature were used after standardization.”

10. In addition to G×E GWAS results, the original GWAS results should also be shown.

Thank you for your comment. The results of GWAS with germination rates at 30 °C and 15 °C are shown in Fig. 1b, c, e, and f.

11. In Extended Data Fig. 2e, the authors claimed that population structure does not have much influence on the data. This would require a more statistical approach as PCA is a visualization approach.

Thank you for your suggestion. we reconducted GWAS with the germination rate at 15 °C with principal components as covariates, greatly improving the appearance of the Q-Q plots (Supplementary Fig. 3c, f). We revised the manuscript and Fig. 1 and Supplementary

Fig.3.

Responses to Reviewer #3 Comments:

1. Genome-wide association study

The authors performed three GWAS tests, including g x e, 30°C, and 15°C. According to Extended data, Fig. 3, g x e and 15°C tests show overinflation of p-values that means failed model fitting. For the linear mixed models like the authors used, the transformation of phenotypes into the most normal (i.e., box-cox) is necessary to reduce false-positive errors from the population structures. The unfitted g x e test seems to be affected by the skewed distribution of 15°C phenotypes.

Thank you for your suggestion. According to your suggestion and those of the other reviewers, we reconducted GWAS with the germination rate at 15 °C transformed by box-cox transformation and principal components as covariates, which greatly suppressed over inflation of Q-Q plots (Supplementary Fig. 2c, f).

2. g x e model

Although the GWAS result is not solid, both 30°C and 15°C (Fig 1b and c) show the same peak around GF14h. It means the peak has common G effects on seed germination rate under both temperatures unless the direction is opposite. Why is the temperature-specific G effect (G x E) so strong in Fig 1a? I think because the common g effect is missing in this G x E model. Usually, Y is defined as the sum of G, E, G x E, Z and error (Z is kinship). Thus, the full model is written as $Y = u + Xb_g + Xb_{g \times e} + Z + e$, and the null model to test $b_{g \times e}$ is $Y = u + Xb_g + Z + e$. If I understand the model in this study (L426 - 432) correctly, the authors used $Y = u + Xb_{g \times e} + Z + e$, and the null model is $Y = u + Z + e$, as they wrote 'G x E is included in X, L429'. In this case, G x E is overestimated because it partially contains common g effects as well.

Ref. i.e.,

Korte et al., Nature genetics (2012)

<https://www.nature.com/articles/ng.2376>

Moore et al., Nature genetics (2019)

<https://www.nature.com/articles/s41588-018-0271-0>

We appreciate this comment. After careful evaluation, we found several statistical problems with our previous GxE GWAS methods. We found that these GxE GWAS problems (score inflation, overestimation of GxE effect, etc.) have recently been discussed by Yamamoto and Matsunaga (G3, 2021). Therefore, we invited Dr. Yamamoto, as a co-author to review our GxE GWAS. The details of the methodology used are described in Methods section and has been added below for your perusal. As a result of this reanalysis, we believe that the Q-Q plots have significantly improved (Supplementary Fig. 3a, d in the revised manuscript), and also resolves the concerns you mentioned above. “The GWAS for each environmental condition was performed based on the following linear mixed model⁴⁹:

$$\mathbf{y} = \mathbf{S}\mathbf{s} + \mathbf{x}\alpha + \mathbf{u}_G + \boldsymbol{\varepsilon} \quad (3)$$

where, \mathbf{y} and $\boldsymbol{\varepsilon}$ indicate $n \times 1$ vectors for phenotypic values and residuals, respectively; \mathbf{S} is an $n \times c$ matrix whose column elements are the eigen vectors from principal component analysis of genotype data from all markers. In this study, eigen vectors from the first five principal components were used. \mathbf{s} indicates $c \times 1$ vector of fixed effects for \mathbf{S} . \mathbf{x} is an $n \times 1$ vector of marker genotype values coded as $\{-1, 1\} = \{AA, aa\}$; α is the marker effect. \mathbf{u}_G models the random effects as follows:

$$\mathbf{u}_G \sim MVN(\mathbf{0}, [\mathbf{Z}_G \mathbf{G} \mathbf{Z}'_G] \sigma_G^2) \quad (4)$$

where MVN is the multivariate normal distribution; and \mathbf{Z}_G represents an $n \times m$ incidence matrix for the phenotype and random effects. The genotype-by-environment ($G \times E$) GWAS was performed based on the recommended method in Yamamoto and Matsunaga (2021)³¹.

$$\mathbf{y} = \mathbf{T}\mathbf{t} + \mathbf{S}\mathbf{s} + \sum_l^L \{(\boldsymbol{\pi}_l \circ \mathbf{x})\zeta_l\} + \mathbf{u}_G + \mathbf{u}_{GE} + \boldsymbol{\varepsilon} \quad (5)$$

where \mathbf{t} is an $n \times 1$ vector containing indicator variables that determine whether the phenotypic value is obtained from l -th environment $\{1\}$ or not $\{0\}$. ζ_l is the marker effect in the l -th environment. \mathbf{u}_{GE} models the $G \times E$ random effects as follows:

$$\mathbf{u}_{GE} \sim MVN(\mathbf{0}, [\mathbf{Z}_G \mathbf{G} \mathbf{Z}'_G] \circ [\mathbf{Z}_E \mathbf{Z}'_E] \sigma_{GE}^2) \quad (6)$$

where \mathbf{Z}_E is the incidence matrix for the phenotypic values and environmental

differences (i.e., 15 °C or 30 °C in this study); and σ_{GE}^2 is the variance for \mathbf{u}_{GE} . The model without marker G × E effect is as follows:

$$\mathbf{y} = \mathbf{Tt} + \mathbf{Ss} + \mathbf{x}\alpha + \mathbf{u}_G + \mathbf{u}_{GE} + \boldsymbol{\varepsilon} \quad (7)$$

The statistical significance of marker G×E effects was evaluated using the log-likelihood (LL) ratio test (LRT). The *P*-value of G × E effect for each marker was calculated using the chi-square test based on the deviance *D*:

$$D = -2 \times (LL_{Eq.5} - LL_{Eq.7}) \quad (8)$$

with the degree of freedom equal to *L*-1 (i.e., 1 in this study). The algorithms to estimate each parameter and to fit the models are given in detail in Yamamoto and Matsunaga (2021)³¹. Manhattan plots and quantile-quantile (Q-Q) plots with $-\log_{10}$ *P*-values analysed by LMM were generated using the R package qqman⁵⁰. Germination rate for 15 °C were used after Box-Cox transformation optimized in terms of normality. For G × E GWAS, the germination rate at each temperature were used after standardization.”

3. Identification of the causality

The peak1 is very wide, over 500kbp. How many genes are involved in the region? Why can authors exclude possibilities that causality affects traits through gene expression regulation? Please clarify whether the authors claim GF14h is only a causal gene in the haplotypes or not. Also, how much phenotypic variation is explained by the peak1 and the 4bp deletion in GF14h?

Thank you for the above comments. Our GWAS panel consists only of temperate *japonica* varieties grown in Japan and, as previously reported, has a very low polymorphism frequency (e.g., Figure 3c and 5a in Yano et al., Nature Genetics, 2016; Figure 2A in Yano et al., PNAS, 2019) and is characterized by a much wider range of LDs compared to many other panels. In addition, there are many genes annotated as "retrotransposon proteins" or "putative proteins" in this region. As a result, as you pointed out, the peak1 region is wide, about 500KB, but the number of candidate polymorphisms in this region is also limited. In the present case, five candidate genes were present within this approximately 500 KB region (Supplementary Fig. 5b).

As shown in Supplementary Table 4, the polymorphisms with the highest $-\log_{10}(p)$ values in Peak 1 and 4 bp Indel were shown to explain 33 % and 15 % of the phenotypic

variance, respectively. This difference is attributed to the difference in missing genotypes between Peak 1 and 4bp Indel. Thus, the percent of variance was inadequate to indicate the effect of a significant signal on the region. Therefore, we calculated the effect sizes (estimated effect on germination rate) of these polymorphisms as pointed out by the reviewer. The effect of these polymorphisms on germination rate was calculated to be 58.6% and 54.4% using previously published methods (Yamamoto and Matsunaga, G3, 2021). Although the standard deviation of the estimated contribution is high, it is reasonable to assume that the estimated effect of the 4 bp deletion is compatible with the difference between the germination tests of NPB and NPB controls with Hap.2 shown in Figure 3e (54.4 % versus 47-60 % at 48 h). These results strongly suggest that the effect of peak1 can be largely explained by the 4 bp deletion, while the other polymorphisms have a minor effect on temperature-dependent germination. These additional statistical results are shown in Supplementary Table 2–4. The manuscript was revised to clarify this as follows (P11, L246-249): “This analysis identified the region on Chr. 11 containing the gene(s) responsible for temperature-dependent germination. The following physiological and biochemical analyses confirmed GF14h as at least one of the $G \times E$ gene, preferentially functioning at 30 °C but not at 15 °C.”

4. Fig 2c-d, The raw values contain effects of population structures. Please use BULP to show the allelic effects.

We wanted to show the performance of each cultivar in Fig 2c–d. Therefore, we have revised the figure legends as follows (P35, L813-816):

“**c, d**, Violin plots showing germination rate of cultivars that have loss-of-function (haplotype [Hap.]1) and gain-of-function (Hap.2/3) at 30 °C for 24 h and 15 °C for 96 h, respectively. .”

In addition, as the reviewer pointed, BLUP estimation of allelic effects must be presented in the manuscript. We also showed the BLUP estimated allelic effects in Supplementary Table 2 and 3 of the revised manuscript.

5. L217, Are the authors suggesting allelic heterogeneity? Instead of “the novel

approach”, the traditional co-factor model works in the same way and that’s more straightforward for the case.

<https://www.nature.com/articles/nature08800>

Actually, analyses using synthesized SNPs based on the functionality are not a novel approach.

Thank you for the suggestion. We tone-downed the description for this approach. To clarify our message, we have revised our description as follows (P5, L91-93): “The most of GWAS platforms are designed to analyse bi-allelic variants, and therefore, multi-allelic variants are ignored or forcibly converted into bi-allelic state (e.g., reference type and another major allele)¹⁰.”

6. What is the reason that the low germination rate of transgenic lines with Hap2 (Fig2e and f) compared with the natural lines (Fig2c and d) at 24h? In addition, if you compare germination rate at 30°C 24h and 15°C 96h, Hap2 genome/NBP might have the same effect in both temperatures.

Seed germination rates greatly depend on its physiological condition. In particular, seeds collected in a field and that in a greenhouse cannot be compared in the same context. In the present case, the germination rate of Hap. 1 plants is 30-40% and that of Hap. 2 plants is 60-70% at 30 °C for 24 hours (Fig. 2c), whereas that of recombinant seeds is very low under the same condition (Fig. 2e). There are two major possible causes for this difference. One is the different seed harvesting conditions (field and greenhouse), and the other is storage period of the seeds (the storage period of the recombinant seeds is shorter than the field harvesting seeds). Such difference in the germination rate between natural and transgenic lines has been often observed (e. g. Fig. 2B vs Fig. 4A,B in Fujino et al., PNAS, 2008). Therefore, when comparing germination rates between seeds collected under different conditions, we did not simply compare germination rates per se, but always compared the germination rates of varieties with Hap.2/Hap.3 against those with Hap.1, as mentioned above in the answer to your #3 comment.

L266, How did the authors estimate the selection pressure? EHH does not take into account the population structure and the sampling bias. If the authors claim the allele is under positive selection, please show and compare the genome-wide significance. Without any population genetic analyses, the selection pressure is speculation (L266).

Thank you for your suggestion. As we agree with your opinion, we removed the figure about EHH and revised it to tone-down about selective pressure.

How is heritability (= genetic effect) under 15°C and 30°C?

We calculated REML estimated narrow-sense heritability (P4, L81-82). The heritabilities were 51.5% and 53.2% for 30 °C and 15 °C, respectively.

Please explain the method to predict the haplotype in materials and methods.

The definition of haplotypes is having unique polymorphisms causing amino acid substitution and more than 2 accessions carrying the haplotype in the population used in this study. We added the explanation in the Methods section (P29, L698-699).

Why can the authors conclude that GF14h in the Hap1 is the loss-of-function gene? There are many cases that the indels do not retrieve the gene function completely.

Thank you for the important suggestion. GF14h^{Hap.1} lacks the conserved region at C-terminus by the 4 bp in-del. The conserved region contains the amino acids important to interact with bZIP and PEBP proteins (Taoka et al., Nature, 2011). To clarify this point, we revised the manuscript (P5, L104-107) and Supplementary Fig. 6.

In addition, we conducted additional experiments demonstrating that Hap.1 could not interact with these proteins *in vivo* and could not regulate transcriptional activity of OREB1 (Supplementary Fig. 10e and 15b).

Responses to Reviewer #4 Comments:

1. I have found that the genetic studies leading to the identification of GF14h alleles (Figs 1 and 2) are well performed and convincing, as well as the experiment demonstrating that GF14h is a negative regulator of ABA signalling. However, in my opinion a fundamental point is lacking: there is no functional demonstration that the function of 14-3-3 in the regulation of germination, emerged from genome-wide association studies, is carried out through the regulation of the transcriptional complex OREB1 / MFT2. In this way, the characterization of the interaction between GF14h, OREB1 and MFT2, although interesting, does not make a real contribution to provide a rationale for the identification of GF14h as an ABA-dependent regulator of seed germination. For example, in principle, GF14h could negatively regulate ABA signalling by activating a physiological process antagonistic to ABA (e.g., 14-3-3 have a role in the regulation of signalling and biosynthesis of GA, which play an opposite role respect to ABA in the germination process). I think that the authors should address this point or at least they should critically discuss in the manuscript.

Thank you for your comments. We performed the following experiments to examine GF14h interaction with OREB1 to participate in ABA signaling and thus inhibit germination. First, compared to NPB with GF14h^{Hap1} (loss-of-function), NPB with GF14h^{Hap2} introduced by transformation showed a clear reduction in the inhibitory effect of ABA on seed germination (Fig. 3a). Furthermore, the expression of ABA responsive genes, *OsRab16A*, *OsLea3*, and *OsEM*, was suppressed in GF14h^{Hap2}-transformed NPB compared to NPB (Fig. 3b). Second, OREB1, one of interactants of GF14h, is known to be a key component of ABA signaling (Skubacz et al., *Front. Plant Sci.*, 2016), and a key regulator of seed germination in the ABA signaling pathway (Li et al., *Front. Plant Sci.*, 2021), which was published after our submission and is cited in the revised version. In this context, the transient reporter assays in rice protoplasts demonstrated that transcriptional activation of OREB1 against ABA-responsive *OsEM* gene was substantially alleviated by GF14h (Fig. 4d, e). According to these results, we concluded that GF14h regulates seed germination through manipulation of ABA signaling pathway.

2. Did the authors verify whether GF14Hap1 is normally expressed in the plant? The

authors should investigate this aspect, in order to understand if the Hap1 allele is loss of function due to a lack of / insufficient expression or because the amino acid substitutions result in a lower ability of GF14Hap1 to interact with target proteins. In the Yeast Two-hybrid Assay, is the absence of β -galactodidase activity due to the lack of expression of the protein in yeast or rather to the lack of ability of GF14 Hap1 to interact with OREB1 and TRAB1? Again, in the protein-protein interaction experiments (e.g. BiFC and Co-IP experiments) did the authors test the properties of GF14 Hap1? The authors should address these points.

Thank you for the comment. There are four independent data showing the presence of GF14h^{Hap.1} mRNA in NPB, as shown in Supplementary Fig. 9.

We believe that the protein produced from GF14h^{Hap.1} has no physiological activity at all for the following reasons. First, it lacks the conserved region at C-terminus by the 4 bp in-del. The missing region contains the amino acids important for protein-protein interactions and are conserved among all the 14-3-3 proteins, not only plants but also mammals and yeast (Taoka et al., Nature, 2011), and it is quite reasonable to assume that the loss of this region would result in the loss of function of this protein. To clarify this point, we revised the manuscript (P5, L104-107) and Supplementary Fig. 6. In addition, we conducted the additional experiments demonstrating that GF14h^{Hap.1} did not interact with OREB1 or MFT2 *in vivo*, and did not affect the transcriptional activity of OREB1 (Supplementary Fig. 10e and 15b). From the results above, we concluded that GF14h^{Hap.1} is a loss-of-function haplotype.

3. In the confocal microscope experiments, the interpretation of the localization of 14-3-3 is not entirely convincing (Fig. 3d, first lane). The nuclear signal, if present, is masked by the cytosolic one. Furthermore, it could also be present a signal at the plasma membrane level. This signal could be relevant for the purposes of the manuscript, as a known function of the 14-3-3 proteins is the regulation of the plasma membrane H⁺-ATPase, which plays an important role in the regulation of seed germination. I strongly suggest to use fluorescent markers for the nucleus and plasma membrane. Minor point: the reference of the scale bar is missing.

Thank you for your important remarks. We reanalyzed the subcellular localization of GF14h^{Hap.2} and found that GF14h^{Hap.2} co-localized with the nucleus markers and not to the plasma membrane or other subcellular compartments markers and dye that we used (Supplementary Fig. 10a–d). Each image is marked with a scale bar.

4. The introduction completely lacks a description of the molecular mechanism of 14-3-3 action. This makes it difficult to understand some experiments, e. g. Figs 3 and 4, where a reader non-expert in the 14-3-3 field may not understand that the S358A mutation prevents phosphorylation and consequently 14-3-3 association to a consensus motif, while the S385E mutation mimics phosphorylation. I strongly suggest to include in the introduction a short introduction on 14-3-3 including the essential bibliography, also totally missing.

Thank you for the important suggestion. According to your suggestion, we added the following sentences with citation of the proper references to explain 14-3-3:

“14-3-3 proteins are highly conserved proteins, widespread in eukaryotic organisms⁷. Among eukaryotes, plants have the largest number of 14-3-3 genes, with 15 in *Arabidopsis* and 8 in rice⁸. The protein family members are classified according to their amino acid sequence similarities into two distinct groups: the ϵ and the non- ϵ group⁸. A common trait of 14-3-3 is their ability to bind to target proteins through the recognition of phosphorylated consensus motifs⁸. Depending on the phosphorylated target, association of 14-3-3 proteins can have different functional consequences, leading to regulation of its enzymatic activity, subcellular localization, protein stability or alteration of protein-protein interactions⁸. At present, several 14-3-3 interactants playing a pivotal role in various physiological processes, such as growth and development and stress response, have been identified⁸. Additionally, a growing body of evidence has emerged regarding the involvement of 14-3-3 proteins as key players in different aspects of plant hormone physiology⁸.”

Reviewers' Comments:

Reviewer #1:

Remarks to the Author:

The manuscript has been significantly improved. Thank you for all the revisions!

Reviewer #2:

Remarks to the Author:

The revised manuscript has been improved and most of questions have been addressed. The functions GF14Hap.2, OREB, and MFT2 involved in ABA-regulated seed germination is clearly elucidated.

However, I have some concerns that would further strengthen the manuscript.

1. In Fig 4i, working model. A previous study showed that high temperature inhibits seed germination by inducing ABA biosynthesis and repressing GA action in Arabidopsis (Toh et al., Plant Physiol. 2008). However, this model showed that low temperature can induce ABA production, whereas high temperature decreases ABA accumulation. The authors should cite related papers and explain this contradiction.

2. Based on the working model, the ABA responding genes should be responsive to low and high temperatures. This point should be further verified.

3. It would be better to describe why they select S385 in OREB1 in this study.

4. The BiFC signal of GF14hHap.2-MFT2 was observed mainly in the ER. How does MFT2 disrupt the inhibition of GF14hHap.2 on OREB1 in the nucleus? The authors should explain it.

Reviewer #3:

Remarks to the Author:

The authors have reconsidered their statistical analyses and added substantial experiments according to the reviewer's comments. I think their effort improved the manuscript, but the statistical part is still not sufficient.

First of all, please go through ms of the GWAS part carefully with a statistics specialist. What is T_t in equations 5 and 7? What are π and L ? All letters must be defined in text clearly. Regarding the log-likelihood test of $g \times e$ in equation 8, which is the NULL model corresponding to the full model? Please explain this in text clearly. If the authors compared equations 5 and 7, all $g \times e$ effects in L environments are tested together as the summation? I don't understand this model.

Again, how many genes are involved in the region?

As you know, GWAS takes advantage of linkage disequilibrium to detect association by using incomplete genotypes, but LD also becomes a disadvantage, making narrowing down the peak harder. Therefore, the significant eight polymorphisms must be tagged with many haplotypes, including other candidate genes. SNPs tagged with multiple alleles often provide higher significance by increased allele frequency in the case there is allelic or genetic heterogeneity. Confirmation by molecular experiments such as reciprocal alleles or genome editing is the only way to prove the real causality. However, it's time-consuming and beyond the scope of this study. Thus, you found a good candidate variant (or alleles), and the statistical result was consistent with the hypothesis, but you cannot conclude that's the real causality so far.

Line 79, L83: "with a reasonable the Q-Q plot" can be omitted because QQ plots must be proper if you show the GWAS results in research articles. Honestly, I'm not sure whether QQ plots in Fig S3a, d are optimized. Estimation of inflation factor λ is helpful to figure out.

Line 91-92: This phenomenon is called allelic heterogeneity. See Atwell et al., 2010,

<https://doi.org/10.1038/nature08800>
Line 94-95: What is one multi-allelic variant?

Reviewer #4:

Remarks to the Author:

In the revised version of this manuscript the authors have made substantial corrections. Most of the issues I raised have been adequately addressed. In my opinion, in this form the manuscript has been greatly improved, and it is suitable for publication in Nature Communications

Reviewer #1 (Remarks to the Author):

The manuscript has been significantly improved. Thank you for all the revisions!

Thank you for your support!

Reviewer #2 (Remarks to the Author):

The revised manuscript has been improved and most of questions have been addressed. The functions GF14Hap.2, OREB, and MFT2 involved in ABA-regulated seed germination is clearly elucidated. However, I have some concerns that would further strengthen the manuscript.

1. In Fig 4i, working model. A previous study showed that high temperature inhibits seed germination by inducing ABA biosynthesis and repressing GA action in Arabidopsis (Toh et al., Plant Physiol. 2008). However, this model showed that low temperature can induce ABA production, whereas high temperature decreases ABA accumulation. The authors should cite related papers and explain this contradiction.

Thank you for the important comment. “High temperature” in the previous manuscript is 30 °C, which is optimum temperature for rice germination as mentioned in Introduction. Therefore, because 30 °C is not stress condition for rice germination, we believe there is no contradiction with the previous reports including that suggested by the reviewer. To avoid this confusion, we replaced “high temperature” with “optimum temperature” in the current manuscript.

2. Based on the working model, the ABA responding genes should be responsive to low and high temperatures. This point should be further verified.

Thank you for the comment. As mentioned above, 30 °C is optimum temperature for rice germination. I apologize to confuse you. About germination under the low temperature, Li et al. (Front. Plant Sci., 2021) have already verified that ABA content and the expression of ABA responding genes, such as OREB1 and the ABA catabolic gene, are responsive to low temperature during germination. We discussed our working model, citing this paper and also referring to these observations.

3. It would be better to describe why they select S385 in OREB1 in this study.

Thank you for the comment. To explain the reason why we used S385 in this study, we have revised the manuscript as follows: “Thus, we replaced S385 in OREB1, which corresponds to the phosphorylation site essential for the OREB1 and GF14 protein interaction¹⁸, with Ala (S385A) or Glu (S385E) to mimic dephosphorylation and phosphorylation, respectively, in order to examine the role of its phosphorylation state.”

4. The BiFC signal of GF14hHap.2-MFT2 was observed mainly in the ER. How does MFT2 disrupt the inhibition of GF14hHap.2 on OREB1 in the nucleus? The authors should explain it.

The purpose of this model is to explain how the transcriptional activity of OREB1 is regulated by GF14h and MFT2. OREB1 (either alone or combined with GF14h and MFT2) is always present in the nucleus. Therefore, even if the GF14h-MFT2 interaction occurs outside the nucleus, this can be an independent event with respect to transcriptional regulation of OREB1. In fact, GF14h-OREB1-MFT2 has been confirmed to be localized in the nucleus (Fig. 4f).

Reviewer #3 (Remarks to the Author):

The authors have reconsidered their statistical analyses and added substantial experiments according to the reviewer's comments. I think their effort improved the manuscript, but the statistical part is still not sufficient.

First of all, please go through ms of the GWAS part carefully with a statistics specialist. What is T_t in equations 5 and 7? What are π and L ? All letters must be defined in text clearly. Regarding the log-likelihood test of $g \times e$ in equation 8, which is the NULL model corresponding to the full model? Please explain this in text clearly. If the authors compared equations 5 and 7, all $g \times e$ effects in L environments are tested together as the summation? I don't understand this model.

Thank you for the important suggestion. We totally revised the method of the GWAS part as follows. Each item pointed out by the reviewer is also described below, with the corresponding

section highlighted in yellow. We described the definition of all of letters in text clearly. “L”, corresponding to the number of environmental conditions, was replaced with “2” which is the actual number used in this study. NULL model is (4) and the full model is (7) described as “the alternative model”. The null hypothesis of the $G \times E$ GWAS in this study is that a marker has a common effect in both experimental conditions (i.e., 15 °C and 30 °C) but does not have an effect specific to each experimental condition and the alternative hypothesis of the $G \times E$ GWAS in this study is that a marker has different effects between the experimental conditions. We compared these equations.

“The GWAS for each environmental condition was performed using the function 'GWAS' in the R package rrBLUP version 4.3 with default parameter settings except for $n.PC=5^{46}$.

This study's genotype-by-environment ($G \times E$) GWAS was performed based on the recommended method in Yamamoto and Matsunaga (2021)³¹. The null hypothesis of the $G \times E$ GWAS in this study is that a marker has a common effect in both experimental conditions (i.e., 15 °C and 30 °C) but does not have an effect specific to each experimental condition. Therefore, the null model is as follows:

$$\mathbf{y} = \mathbf{T}\mathbf{t} + \mathbf{S}\mathbf{s} + \mathbf{x}\alpha + \mathbf{u}_G + \mathbf{u}_{GE} + \boldsymbol{\varepsilon} \quad (4)$$

where \mathbf{y} and $\boldsymbol{\varepsilon}$ indicate $n \times 1$ vectors for phenotypic values and residuals, respectively; n is the number of phenotypic records; \mathbf{T} is an $n \times 2$ design matrix that assigns phenotypic values to the experimental conditions, \mathbf{t} is a 2×1 vector of the population-wide mean for each experimental condition; \mathbf{S} is an $n \times 5$ matrix whose column elements are the first five eigenvectors from principal component analysis of genotype data from all markers. \mathbf{s} indicates a 5×1 vector of fixed effects for \mathbf{S} . \mathbf{x} is an $n \times 1$ vector of genotype values of a marker coded as $\{-1, 1\} = \{\text{REF/REF}, \text{ALT/ALT}\}$; α is a marker effect common to both experimental conditions. \mathbf{u}_G models the random effects common to both experimental conditions:

$$\mathbf{u}_G \sim MVN(\mathbf{0}, [\mathbf{Z}_G \mathbf{G} \mathbf{Z}'_G] \sigma_G^2) \quad (5)$$

where MVN is the multivariate normal distribution; \mathbf{Z}_G represents an $n \times m$ incidence matrix for the phenotype and random effects; m is the number of varieties (i.e., $m = 164$ in this study); \mathbf{G} is the $m \times m$ genetic relationship matrix calculated by function 'A.mat' in the R package rrBLUP version 4.3^{46, 47}; σ_G^2 is the variance for \mathbf{u}_G . \mathbf{u}_{GE} models the $G \times E$ random effects as follows:

$$\mathbf{u}_{GE} \sim MVN(\mathbf{0}, [\mathbf{Z}_G \mathbf{G} \mathbf{Z}'_G] \circ [\mathbf{Z}_E \mathbf{Z}'_E] \sigma_{GE}^2) \quad (6)$$

where \mathbf{Z}_E is the $n \times 2$ incidence matrix for the phenotypic values and environmental differences (i.e., 15 °C or 30 °C in this study); σ_{GE}^2 is the variance for \mathbf{u}_{GE} ; the symbol \circ indicates the Hadamard product for the left and right vectors or matrices. The alternative hypothesis of the $G \times E$ GWAS in this study is that a marker has different effects between the experimental conditions. Therefore, the alternative model is as follows:

$$\mathbf{y} = \mathbf{T}\mathbf{t} + \mathbf{S}\mathbf{s} + \sum_l^2 \{(\boldsymbol{\pi}_l \circ \mathbf{x}) \zeta_l\} + \mathbf{u}_G + \mathbf{u}_{GE} + \boldsymbol{\varepsilon} \quad (7)$$

where π_l is an $n \times 1$ vector containing indicator variables that determines whether the phenotypic value is obtained from the l -th experimental conditions {1} or not {0}; and ζ_l is the marker effect in the l -th experimental conditions. The statistical significance of marker G×E effects was evaluated using the log-likelihood (LL) ratio test (LRT). The P -value of the G × E effect for each marker was calculated using the chi-square test based on the deviance D :

$$D = -2 \times (LL_{Eq.7} - LL_{Eq.4}) \quad (8)$$

with the degree of freedom equal to 1. The theoretical validity and algorithms to estimate each parameter and fit the models are detailed in Yamamoto and Matsunaga (2021)³¹.

Again, how many genes are involved in the region?

As you know, GWAS takes advantage of linkage disequilibrium to detect association by using incomplete genotypes, but LD also becomes a disadvantage, making narrowing down the peak harder. Therefore, the significant eight polymorphisms must be tagged with many haplotypes, including other candidate genes. SNPs tagged with multiple alleles often provide higher significance by increased allele frequency in the case there is allelic or genetic heterogeneity. Confirmation by molecular experiments such as reciprocal alleles or genome editing is the only way to prove the real causality. However, it's time-consuming and beyond the scope of this study. Thus, you found a good candidate variant (or alleles), and the statistical result was consistent with the hypothesis, but you cannot conclude that's the real causality so far.

Thank you for the comment. Based on the reviewers' comments, we examined the four possible candidates other than GF14h listed in Supplementary Fig. 5b and concluded that none of them are likely to be involved in regulating germination for the following reasons. First, these genes were found no or trace level of expression in seeds; second, the amino acid residues varied by the polymorphisms are highly variable among their homologous proteins, making it very unlikely that such amino acid variations would affect protein function. Actually, LOC_Os11g39530/Os11g0609500 and LOC_Os11g39640/Os11g0610600 can be aligned with their orthologs in grass plants. The amino acid substitutions caused by the SNPs with high P values occurred only within non-conserved regions within the *Oryza* genus, and thus it is unlikely that these mutations affect the gene function. Furthermore, little expression of these two genes was observed in seeds, strongly suggesting that they are not involved in seed germination. LOC_Os11g39680/Os11g0611100 has no homologs in plants outside the *Oryza* genus, and A (amino acid of the reference genome) and V substituted by the SNPs with high P values are coexisting within the *Oryza* genus, suggesting that this amino acid replacement is a neutral mutation within the genus. Furthermore, this gene is not expressed at all in seeds and thus very

unlikely to be involved in germination regulation. LOC_Os11g39800 is predicted as a gene in MSU Rice Genome Annotation Project (MSU; <http://rice.uga.edu/>) but not in Rice Annotation Project Database (<https://rapdb.dna.affrc.go.jp/>). Furthermore, Phytozome v13 (<https://phytozome-next.jgi.doe.gov/>) does not predict any homologs for this gene, and MSU database states that this gene expression is absent in all organs. Thus, we also removed this gene as a candidate. These additional results are presented in the revised manuscript as Supplementary Fig. 6, and we revised the manuscript as follows: “When evaluating the functional impact of these mutations (Supplementary Fig. 6 and 7), we focused on a 4 bp InDel in the coding sequence of LOC_Os11g39540/Os11g0609600, which was annotated as a 14-3-3 protein, GF14h (Fig. 2b)¹².”

Line 79, L83: "with a reasonable the Q-Q plot" can be omitted because QQ plots must be proper if you show the GWAS results in research articles. Honestly, I'm not sure whether QQ plots in Fig S3a, d are optimized. Estimation of inflation factor lambda is helpful to figure out.

Thank you for your suggestion. According to your suggestion, we omitted “with a reasonable the Q-Q plot”.

Line 91-92: This phenomenon is called allelic heterogeneity. See Atwell et al., 2010, <https://doi.org/10.1038/nature08800>

Line 94-95: What is one multi-allelic variant?

Thank you for the comments. According to your suggestion, we revised the manuscript related to allelic heterogeneity as follows: “The most of GWAS platforms are designed to analyse bi-allelic variants, and therefore, allelic heterogeneity¹⁰) are ignored or forcibly converted into bi-allelic state (e.g., reference type and another major allele)¹¹. We speculated that these functionally different alleles of *qLTG3* may decrease the statistical power of the GWAS. In the variant list, these polymorphisms were regarded as allelic heterogeneity and neglected for GWAS.”

Reviewer #4 (Remarks to the Author):

In the revised version of this manuscript the authors have made substantial corrections. Most of the issues I raised have been adequately addressed. In my opinion, in this form the manuscript has been greatly improved, and it is suitable for publication in Nature Communications

Thank you for your support!

Reviewers' Comments:

Reviewer #2:

Remarks to the Author:

The authors have nicely adjusted the manuscript. I have no further comments.

Reviewer #3:

Remarks to the Author:

I appreciate the effort of the authors in revising the manuscript. The article was sufficiently improved, and the statistical part looks okey. I left a couple of comments.

Regarding the response, "We described the definition of all of letters in text clearly. "L", corresponding to the number of environmental conditions, was replaced with "2" which is the actual number used in this study."

I understand, but the magic number should not be used without any explanation in the statistical part, and "2" would be unclear to readers what is indicated. Can you edit the sentence like that?

" τ is an $n \times 2$ design matrix that assigns phenotypic values to the two experimental conditions, 15C and 30C, and t is a 2×1 vector of the population-wide mean for each experimental condition."

There are many established $G \times E$ GWAS models. If the authors consider other models next time, the analyses might become more straightforward and solid.

GEMMA <https://www.nature.com/articles/nmeth.2848>

LIMIX <https://europepmc.org/article/PPR/ppr7019>

MTMM <https://pubmed.ncbi.nlm.nih.gov/22902788/>

"Based on the reviewers' comments, we examined the four possible candidates other than GF14h listed in Supplementary Fig. 5b and concluded that none of them are likely to be involved in regulating germination for the following reasons".

I was just asking how many genes are involved in the region because non-synonymous SNPs are not only the causality of the phenotypic variation. Molecular biologists tend to consider natural variation as a kind of "natural mutants", but the idea is wrong.

Reviewer #2 (Remarks to the Author):

The authors have nicely adjusted the manuscript. I have no further comments.

Thank you for your support!

Reviewer #3 (Remarks to the Author):

I appreciate the effort of the authors in revising the manuscript. The article was sufficiently improved, and the statistical part looks okay. I left a couple of comments.

Regarding the response, “We described the definition of all of letters in text clearly. “L”, corresponding to the number of environmental conditions, was replaced with “2” which is the actual number used in this study.”

I understand, but the magic number should not be used without any explanation in the statistical part, and “2” would be unclear to readers what is indicated. Can you edit the sentence like that?

“ τ is an $n \times 2$ design matrix that assigns phenotypic values to the two experimental conditions, 15C and 30C, and t is a 2×1 vector of the population-wide mean for each experimental condition.”

There are many established G x E GWAS models. If the authors consider other models next time, the analyses might become more straightforward and solid.

GEMMA <https://www.nature.com/articles/nmeth.2848>

LIMIX <https://europepmc.org/article/PPR/ppr7019>

MTMM <https://pubmed.ncbi.nlm.nih.gov/22902788/>

Thank you for your suggestion. We revised the method section accordingly.

“Based on the reviewers' comments, we examined the four possible candidates other than GF14h listed in Supplementary Fig. 5b and concluded that none of them are likely to be involved in regulating germination for the following reasons”.

I was just asking how many genes are involved in the region because non-synonymous SNPs are not

only the causality of the phenotypic variation. Molecular biologists tend to consider natural variation as a kind of “natural mutants”, but the idea is wrong.

Thank you for the comment. We understand the reviewer’s concern and have already stated “The following physiological and biochemical analyses confirmed GF14h as at least one of the $G \times E$ gene, preferentially functioning at 30 °C but not at 15 °C.” in the manuscript. Therefore, we believe there is no need to revise any sentences in response to this comment.